# scCMIA: Self-supervised Dual Model for Mitigating Information Loss in Single-cell Cross-Modal Alignment

## Abstract

Recent technological advances in single-cell sequencing have enabled simultaneous profiling of multiple omics modalities within individual cells. Despite these advancements, challenges such as high noise levels and information loss during computational integration persist. While existing methods align different modalities, they often struggle to balance alignment accuracy with the preservation of modality-specific information needed for downstream biological discovery. In this paper, we introduce scCMIA, a novel framework guided by Mutual Information (MI) principles that leverages a VQ-VAE architecture. scCMIA achieves robust cross-modal alignment in a unified discrete latent space while enabling high-fidelity reconstruction of the original data modalities. Crucially, our framework transforms the learned discrete representations into a tool for tangible biological discovery, allowing for the quantification of regulatory programs and cross-modal relationships. Our extensive experiments demonstrate that scCMIA achieves state-of-the-art performance across multiple datasets. Our code is available at: https://anonymous.4open.science/r/scCMIA-77E3.

## 1 Introduction

Multimodal learning is becoming increasingly crucial in the field of biology. Biological processes within cells involve multiple regulatory levels, including DNA, RNA, and proteins Tang et al. (2023); Li et al. (2024). The intricate interactions and influences between these levels necessitate an integrated multimodal understanding to fully comprehend these biological processes Tu et al. (2022). In recent years, technological advancements that enable the analysis of multimodal information at single-cell resolution have been pivotal in cataloging cell types and states. For instance, single-cell RNA sequencing (scRNA-seq) Picelli et al. (2013) is used to profile the transcriptomes of individual cells, offering deep insights into cellular heterogeneity and gene expression patterns. Similarly, the single-cell assay for transposase-accessible chromatin with high throughput sequencing (scATAC-seq) Cusanovich et al. (2015) profiles the chromatin accessibility of individual cells, providing valuable information about gene regulatory networks and chromatin structure.

Although single-modality data are easy to obtain and analyze, they provide limited insight into how multiple regulatory layers interact within a single cell Wu et al. (2021). Advances in multimodal protocols Chen et al. (2019b); Ma et al. (2020); Xu et al. (2022) now enable simultaneous profiling of gene expression and chromatin accessibility. However, integration remains difficult because the respective feature spaces differ substantially Chen et al. (2019a): scATAC-seq peak accessibility and scRNA-seq gene expression exhibit extreme sparsity, high dimensionality, and intricate many-to-many regulatory mappings between peaks and genes Argelaguet et al. (2021), hindering effective joint analysis. Intuitively, a direct strategy for managing multimodal data is to embed heterogeneous modalities into a unified representation space. Several methods pursue this by mapping multimodal inputs into a shared latent, often leveraging prior knowledge Duren et al. (2018); Zeng et al. (2019). Conversely, alternative approaches—such as Minoura et al. (2021); Gong et al. (2021)—jointly model multiple modalities without enforcing an explicit shared space. While effective for reconstruction and per-modality objectives, such designs may be less suited to tasks that require direct cross-modal comparison (e.g., cell-type querying or label transfer), and—absent a unified representation—can make it harder to fully leverage complementary information across modalities. Furthermore, some

techniques Cao & Gao (2022); Ashuach et al. (2023) combine cross-modal alignment with modality reconstruction to boost performance; nevertheless, there remains room to improve both alignment efficacy and reconstruction fidelity, particularly under sparse, high-dimensional regimes and distribution shifts.

To address these challenges, we developed a novel VQ-VAE-based cell-level alignment framework, called single-cell cross-modal mutual information (MI) alignment (scCMIA). This framework effectively achieves cross-modal alignment at the single-cell level and reconstructs data in their original modality spaces. scCMIA utilizes the RNA to ATAC (RtA) module to initially align scRNA and scATAC sequences in a continuous shared feature space. Subsequently, it constructs a cross-modal unified codebook in discrete space, facilitating enhanced cross-modal interaction and significantly improving the robustness of both alignment and reconstruction processes. This approach not only enhances the alignment accuracy but also effectively reconstruct multi-modal data, thereby addressing the issue of unimodal information deficiency and providing a comprehensive solution for multi-modal data integration and analysis. Our main contributions are summarized as follows:

- We propose scCMIA, a single-cell multi-omics integration framework centered on mutual information (MI) theory. This framework achieves alignment by maximizing cross-modal MI while minimizing intra-modal MI for feature decoupling, providing theoretical assurance for high-precision alignment and high-fidelity data generation. To this end, we designed a robust dual-space alignment strategy that first performs alignment in a continuous space and then refines it in a discrete space using a unified discrete codebook. This approach significantly enhances the model's overall performance and robustness.

- We demonstrate that the designed unified codebook can learn structures with high biological significance, and propose methods to quantify the conservation of regulatory programs across cell lineages and reveal differences in regulatory coupling among distinct cell types, providing powerful new tools for downstream biological exploration.

- Through extensive experiments across multiple benchmark datasets, we demonstrate that scCMIA achieves state-of-the-art performance across a range of key tasks. Compared to existing methods, our model exhibits significant advantages in cross-modal alignment, data reconstruction, and data interpolation tasks, comprehensively validating the effectiveness and superiority of our proposed framework.

## 2 RELATED WORK

Multimodal alignment is rapidly advancing in fields such as text, vision, and speech. Methods like CLIP Radford et al. (2021), ALBEF Li et al. (2021), and GLIP Li et al. (2022) have played significant roles in their respective domains. Concurrently, in the field of biology, multimodal alignment and reconstruction methods are also making important contributions. In the field of biology, we can categorize multimodal integration strategies into three types: (1). multimodal alignment, (2). multimodal reconstruction, and (3). multimodal alignment and reconstruction.

**Multimodal Alignment**. Techniques such as Pamona Cao et al. (2022), UnionCon Cao et al. (2020), Seurat V3 Stuart et al. (2019), MMD_MA Singh et al. (2020), SCOT Demetci et al. (2020) and scGCL Xiong et al. (2023) align cells from different omics layers through nonlinear flows. These methods eliminate the need for prior knowledge and minimize information loss between modalities. However, they suffer from poor alignment robustness when handling noisy and difficult to apply on large scale data processing.

**Multimodal Reconstruction**. Techniques such as scMM Minoura et al. (2021), Cobolt Gong et al. (2021) and scButterfly Cao et al. (2024) focus primarily on reconstructing missing or incomplete data across different modalities. However, as these methods do not explicitly align modalities into a shared latent space, while powerful for reconstruction, they often do not produce a shared latent space. This can pose challenges for tasks like direct cross-modal querying or nearest-neighbor label transfer, which are more naturally performed within a unified embedding.

**Multimodal Alignment and Reconstruction**. Most methods in this category are based on autoencoders. These approaches not only align data from different modalities but also reconstruct the original input data from the aligned representations. GLUE Cao & Gao (2022) utilizes graph variational autoencoders (VAE) to model known regulatory relationships between open chromatin

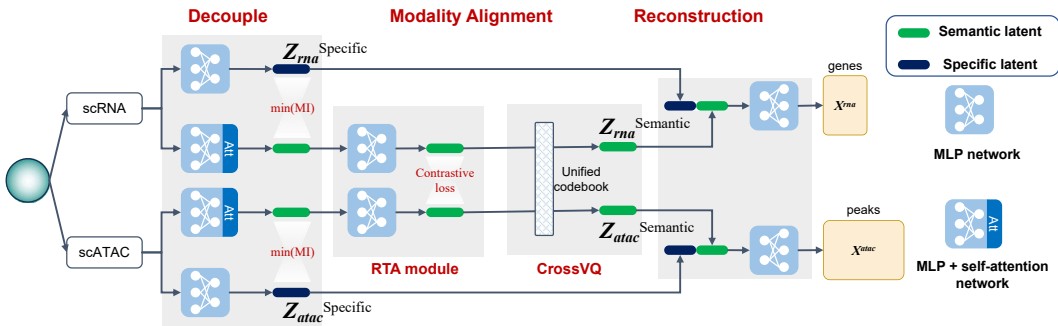

Figure 1: The pipeline of scCMIA. The scCMIA framework is designed to perform intra-modal decoupling and cross-modal alignment, thereby enabling dynamic interaction between modalities while reconstructing individual modalities to capture their intrinsic semantic information.

regions and genes, enabling efficient cross-modal feature translation. However, it can only reconstruct the original spatial data of the scATAC that is related to the scRNA. MutiVI constructs multiple VAE models employing a joint latent representation to integration embedding spaces Ashuach et al. (2023). The performance on alignment and refactoring tasks still has room for further improvement.

**Feature Decoupling**. Feature decoupling plays an important role in many fields, such as Peng et al. (2019) proposed decoupled representation learning framework for multigraphs to capture complete and clean common information. The decoupling of class-independent features is proposed Mo et al. (2023) and the alignment of source domain and target domain is realized. Uni-code Xia et al. (2024) proposed dual cross-modal information uncoupling and multimodal EMA, which unified expression in audio and video, audio text, and even audio - video - text three modes, and realized cross-modal generalization of various tasks in the downstream. Our approach has similar ideas to Uni-code, but we have adopted a completely different strategy in terms of framework and unified codebook design, making it more suitable for the single-cell multi-omics field.

## 3 METHOD

### 3.1 PRELIMINARY

In this section, we first introduce preliminary work on the design of the scCMIA framework (Fig. 1). Our objective is to balance the model in terms of alignment and reconstruction performance. We posit that, for each observed modality $X$ (scRNA) and $Y$ (scATAC), the representation can be decoupled into a modality-agnostic semantic latent $Z^{\text{sem}}$ that captures shared cellular state and a modality-specific latent $Z^{\text{spec}}$ that captures modality-specific signals and technical noise. This decoupling assumption explains and motivates the design choices that follow: (i) minimizing an upper bound on the intra-modality mutual information between $Z^{\text{sem}}$ and $Z^{\text{spec}}$ encourages separation of shared and non-shared factors, (ii) maximizing a lower bound on the cross-modality mutual information between $Z_X^{\text{sem}}$ and $Z_Y^{\text{sem}}$ promotes robust alignment of shared biology across modalities, and (iii) constructing a unified discrete codebook further improves interpretability and cross-modal querying. To operationalize this, we leverage mutual-information (MI) bounds to manage modality-specific features while preserving semantic consistency across modalities.

MI is a measure of mutual dependence between two random variables. Intuitively, MI represents the amount of information contained in one random variable about another. Given random variables $X$ and $Y$, the MI is defined as the Kullback–Leibler (KL) divergence between its joint distribution $p(x, y)$ and the product $p(x)\,p(y)$ of the marginal distributions:

$$\mathrm{I}(X; Y) = D_{\mathrm{KL}}\big(p(x, y) \,\|\, p(x)\,p(y)\big) = \mathbb{E}_{p(x,y)}\left[\log \frac{p(x, y)}{p(x)\,p(y)}\right]$$

$$= \mathrm{H}(Y) - \mathrm{H}(Y \mid X) = \mathrm{H}(X) - \mathrm{H}(X \mid Y).$$

$$(1)$$

where $\mathrm{H}(X)$ and $\mathrm{H}(Y)$ are marginal entropies, and $\mathrm{H}(X \mid Y)$ and $\mathrm{H}(Y \mid X)$ are conditional entropies. When $X$ and $Y$ correspond to each other, $\mathrm{I}(X;Y) = \mathrm{H}(X) = \mathrm{H}(Y)$; when $X$ and $Y$ are independent of each other, $\mathrm{I}(X;Y) = 0$.

In single-cell multi-omics with scRNA and scATAC, we denote the observed modalities by $X$ (scRNA) and $Y$ (scATAC). For each modality, we define modality-agnostic semantic latents $Z_X^{\mathrm{sem}}$ and $Z_Y^{\mathrm{sem}}$, and modality-specific latents $Z_X^{\mathrm{spec}}$ and $Z_Y^{\mathrm{spec}}$. The intra-modal mutual-information term is

$$\mathcal{L}_{\mathrm{intra}} = \mathrm{I}\big(Z_X^{\mathrm{sem}};\, Z_X^{\mathrm{spec}}\big) + \mathrm{I}\big(Z_Y^{\mathrm{sem}};\, Z_Y^{\mathrm{spec}}\big). \tag{2}$$

To achieve the decoupling of modality-specific and semantic features, our objective is to minimize an upper bound surrogate of this quantity. However, minimizing intra-modality MI $\mathrm{I}(Z^{\mathrm{sem}}; Z^{\mathrm{spec}})$ alone is insufficient because, without guidance, the model lacks a criterion for allocating shared versus non-shared information. We therefore introduce an inter-modality alignment objective (maximizing $\mathrm{I}(Z_X^{\mathrm{sem}}; Z_Y^{\mathrm{sem}})$) to direct the model to place shared biological signal into $Z^{\mathrm{sem}}$ while retaining non-shared, modality-specific variation in $Z^{\mathrm{spec}}$.

Therefore, our overall optimization objective combines minimizing the upper bound of MI within modalities to achieve intra-modal decoupling, and maximizing a lower bound of MI across modalities to achieve cross-modal semantic alignment ($\psi$ is the decoupling module parameter):

$$\min_{\psi}\ \mathcal{L}_{\mathrm{intra}} - \mathcal{L}_{\mathrm{inter}}. \tag{3}$$

## 3.2 INTRA-MODAL DECOUPLING LEARNING

When two modes are very different, forcing the two modes to align directly will lead to semantic information loss in the shared space of each mode Niu et al. (2024). To mitigate this tradeoff between alignment and modal information loss, we formalize the decoupling assumption for paired sequencing data $X$ (scRNA) and $Y$ (scATAC) from the same single cell. With $Z^{\mathrm{sem}}$ and $Z^{\mathrm{spec}}$ defined as above, we require:

$$\max\big(\mathrm{I}\big(Z_X^{\mathrm{sem}}, Z_X^{\mathrm{spec}}\big),\ \mathrm{I}\big(Z_Y^{\mathrm{sem}}, Z_Y^{\mathrm{spec}}\big)\big) < \mathrm{I}\big(Z_X^{\mathrm{sem}}, Z_Y^{\mathrm{sem}}\big). \tag{4}$$

On the basis of this assumption, we use Contrastive Log-ratio Upper Bound (CLUB) Cheng et al. (2020) to estimate an upper bound of the intra-modality MI. In the single-modality case for generic variables $(U, V)$, CLUB is defined as:

$$\mathrm{I}_{\mathrm{vCLUB}}(U; V) := \mathbb{E}_{p(u,v)}[\log q_\theta(v \mid u)] - \mathbb{E}_{p(u)}\mathbb{E}_{p(v)}[\log q_\theta(v \mid u)], \tag{5}$$

where $q_\theta(v \mid u)$ is a variational distribution with parameter $\theta$ to approximate $p(v \mid u)$. To achieve the decoupling of scRNA and scATAC modality-specific and semantic representations, we construct two dual encoders per modality (as shown in Fig. 1) and optimize the MI upper bound between the modality-specific representation and the semantic representation. A minibatch estimator for modality $s \in \{X, Y\}$ is:

$$\hat{\mathrm{I}}_{\mathrm{vCLUB}}^{(s)} = \frac{1}{N^2} \sum_{i=1}^{N} \sum_{j=1}^{N} \Big[ \log q_{\theta_s}\big(z_i^{\mathrm{spec},(s)} \mid z_i^{\mathrm{sem},(s)}\big) - \log q_{\theta_s}\big(z_j^{\mathrm{spec},(s)} \mid z_i^{\mathrm{sem},(s)}\big) \Big]. \tag{6}$$

However, it is difficult to obtain meaningful semantic representations for the semantic encoder by relying solely on this module. Therefore, according to the previous assumption, cross-modal MI should be maximized, as higher cross-modal MI helps the CLUB module achieve better decoupling.

## 3.3 INTER-MODAL CONTRASTIVE LEARNING

The **RtA** module maximizes a lower bound on the MI between the semantic representations of an scRNA–scATAC pair Oord et al. (2018b); Poole et al. (2019b). This is achieved by symmetrically

calculating a contrastive loss from the scRNA-to-scATAC direction and the scATAC-to-scRNA direction. Formally, let $Z_X^{\text{sem}}$ and $Z_Y^{\text{sem}}$ be the semantic representations output by the scRNA and scATAC semantic encoders, respectively. We aim to maximize $\text{I}(Z_X^{\text{sem}}; Z_Y^{\text{sem}})$ by minimizing the following symmetric InfoNCE loss:

$$\mathcal{L}_{\text{RtA}} = -\frac{1}{2}\,\mathbb{E}\left[\log\frac{\exp\left(Z_X^{\text{sem}}\cdot Z_Y^{\text{sem}}/\tau\right)}{\sum_{m=1}^{M}\exp\left(Z_X^{\text{sem}}\cdot Z_{Y,m}^{\text{sem}}/\tau\right)} + \log\frac{\exp\left(Z_Y^{\text{sem}}\cdot Z_X^{\text{sem}}/\tau\right)}{\sum_{m=1}^{M}\exp\left(Z_Y^{\text{sem}}\cdot Z_{X,m}^{\text{sem}}/\tau\right)}\right], \quad (7)$$

where $\tau$ is the temperature coefficient, and in-batch negatives are drawn from other items in the batch. Minimizing $\mathcal{L}_{\text{RtA}}$ is equivalent to maximizing $\mathcal{L}_{\text{inter}}$. Thus, the RtA module can be treated as two perspectives on the semantic representation of scRNA–scATAC pairs, which can be maximized by training the RtA module to maximize $\text{I}(Z_X^{\text{sem}}; Z_Y^{\text{sem}})$.

### 3.4 CROSS-MODALITY UNIFIED CODEBOOK

Properties such as scRNA and scATAC, which are inherently discrete data types and possess characteristics somewhat inconsistent with Gaussian assumptions, present a challenge when it comes to understanding cellular heterogeneity quantitatively. This is especially true given that the potential embeddings generated by existing methods are often continuous and may lack direct biological significance Cui et al. (2024). Therefore, to improve the performance of the model in different downstream tasks, we construct a cross-modal discrete unified codebook. Inspired by SimVQ Zhu et al. (2024), we add a simple and efficient linear transformation to the codebook of VQ to accelerate convergence and improve code utilization, and we design Cross-modal VQ (CrossVQ).

CrossVQ first initializes a cross-modal shared codebook $E = \{e_1, e_2, \ldots, e_K\}$, along with a learnable weight matrix $W$. For modality $s \in \{X, Y\}$, let $Z_s^{\text{sem}}$ be the input to VQ and $Z_s^q$ its quantized output. Using the stop-gradient operator $\text{sg}[\cdot]$, we define:

$$Z_s^q = Z_s^{\text{sem}} + \text{sg}\left[q_s W - Z_s^{\text{sem}}\right],$$
$$q_s = \arg\min_{e\in E}\left\|Z_s^{\text{sem}} - eW\right\|. \quad (8)$$

To update the codebook via $W$ for better cross-modal alignment, focusing on $s = X$ (the case $s = Y$ is analogous), we use:

$$\mathcal{L}_{\text{VQ}}^{(X)} = \left\|q_X\ W - \text{sg}\left[Z_X^{\text{sem}}\right]\right\|^2 + \beta\left\|q_X\ W - \text{sg}\left[Z_Y^{\text{sem}}\right]\right\|^2, \quad (9)$$

and the encoder commitment loss:

$$\mathcal{L}_{\text{encoder}}^{(X)} = \frac{\beta}{2}\left\|Z_X^{\text{sem}} - \text{sg}\left[q_X\ W\right]\right\|^2. \quad (10)$$

Here, $\beta$ is used to weight the codebook and encoder loss terms, respectively. For scATAC in cross-modal settings, the losses are analogous. Therefore, the overall loss term for CrossVQ can be expressed as:

$$\mathcal{L}_{\text{CrossVQ}} = \mathcal{L}_{\text{encoder}}^m + \mathcal{L}_{\text{VQ}}^m, \quad m \in \{\text{RNA}, \text{ATAC}\}. \quad (11)$$

### 3.5 OVERALL TRAINING OBJECTIVE

scCMIA is divided into three main components: decoupling, cross-modal alignment, and reconstruction of the original space data. The scCMIA framework is shown in Fig. 1. First, intra-modal minimization of the mutual-information upper bound (Eq. 6) is performed to achieve effective decoupling. Next, RtA and CrossVQ are used for cross-modal alignment in continuous (Eq. 7) and

discrete spaces (Eq. 11), facilitating efficient interaction between modalities. Finally, each modality is efficiently reconstructed in the original space.

$$\mathcal{L}_{\text{scCMIA}} = \hat{\text{I}}_{\text{vCLUB}} + \mathcal{L}_{\text{RtA}} + \mathcal{L}_{\text{CrossVQ}} + \mathcal{L}_{\text{rec}}. \qquad (12)$$

Here $\mathcal{L}_{\text{rec}}$ represents the reconstruction loss for each modality.

## 4 EXPERIMENTS

In this section, we systematically evaluate the performance, robustness, and utility of scCMIA. We begin by describing the diverse benchmark datasets and evaluation metrics used for our comprehensive assessment (Section 4.1). We then demonstrate the superiority of our framework on the primary tasks of cross-modal alignment, assessing both cell matching accuracy and label transfer fidelity (Section 4.2), and high-fidelity data reconstruction (Section 4.3). Subsequently, we validate the core mechanics of our model, presenting quantitative proof of our feature decoupling effectiveness and the necessity of each component via ablation studies (Section 4.4). We further assess the practical utility of our learned embeddings in downstream biological tasks through rigorous cell-type classification on large-scale and CITE-seq data (Section 4.5). Finally, to provide a comprehensive validation, we present critical supplementary experiments in the Appendix, which include imputation accuracy via masking experiments (Appendix B.1), a deep exploration of the biological significance of our unified codebook (Appendix B.1.1), and a rigorous cross-dataset zero-shot generalization test (Appendix B.1.4).

### 4.1 DATASETS AND METRICS

**Datasets**   Single-cell multi-omics data are often hindered by complex and sophisticated techniques, low throughput, and high noise levels. Therefore, in this paper, we use well-studied single-cell multimodal data from the community for testing purposes. Including 10x Multiome PBMC Genomics (2020), SHARE-seq Ma et al. (2020), SNARE-seq Chen et al. (2019b), ISSAAC-seq Xu et al. (2022) and large-scale data 10×BMMC (69,249 cells, 13 batches, and 22 cell types). In addition, we used the CITE dataset of scRNA+ADT, which contains 90,261 samples. Detailed information on these datasets is shown in Appendix Table 9.

**Evaluation Metrics**   The fraction of samples closer than the true match (**FOSCTTM**) Singh et al. (2020) was used to assess the accuracy of the alignment of the single cell level. A lower FOSCTTM value indicates a higher accuracy in correctly identifying that two modalities originate from the same cell. Additionally, we use Root Mean Square Error (**RMSE**) and Mean Absolute Error (**MAE**) to evaluate the reconstruction performance of the model, of which the lower value indicates a better reconstruction performance. Finally, we also verify that the representations obtained from the latent space contain cell identity information using several clustering metrics including Adjusted Rand Index (**ARI**), Normalised Mutual Information (**NMI**), Adjusted Mutual Information (**AMI**) and the **Homogeneity (HOM)** metric items. Classification task uses four core metrics Liu et al. (2025): Overall Accuracy (**OAC**), Average Accuracy (**AAC**) (macro-averaged per-cell-type accuracy, robust to class imbalance), **F1-score** (class-wise harmonic mean of precision and recall, macro-averaged), and **Specificity**.

### 4.2 ALIGNMENT EXPERIMENTS

The core objective of multi-modal alignment is to precisely match corresponding cells across modalities like scRNA-seq and scATAC-seq, a task challenged by their inherently heterogeneous data distributions. To systematically evaluate how different alignment strategies preserve this biological correspondence, we conducted extensive comparative experiments, with quantitative results detailed in Table 1.

The results of the multimodal alignment experiment show that our method achieved the best performance in most datasets, and on average, it reduced the error of the best competing alignment method by 26.60%. To evaluate cross-modal label transferability between scRNA-seq and scATAC-seq data, we performed bidirectional cell type annotation transfer experiments. Specifically, we trained

Table 1: Alignment performance measured by FOSCTTM (mean ± std, lower is better ↓). **Bold** = best, *italic* = second best. Average is mean across non-NA datasets.

| Method | 10X Multiome | ISSAAC-seq | SHARE-seq | SNARE-seq | Average |
|--------|--------------|------------|-----------|-----------|---------|
| MultiVI | 0.2482±0.092 | 0.3679±0.01 | 0.1989±0.003 | 0.2567±0.008 | 0.2346 |
| Seurat v3 | 0.0777±0.0002 | 0.0778±0.0002 | 0.1214±0.001 | 0.2501±0.0004 | 0.1318 |
| GLUE | *0.0172±0.002* | *0.0111±0.002* | *0.0343±0.003* | **0.0127±0.006** | 0.0188 |
| MMD-MA | 0.2998±0.014 | 0.4027±0.049 | — | 0.5280±0.0138 | 0.4102 |
| Pamona | 0.4968 | 0.5007 | — | 0.5025 | 0.5000 |
| UnionCom | 0.5041±0.029 | 0.4875±0.059 | — | 0.4800±0.027 | 0.4905 |
| **scCMIA** | **0.0132±0.008** | **0.0027±0.001** | **0.0165±0.003** | *0.0227±0.006* | **0.0138** |

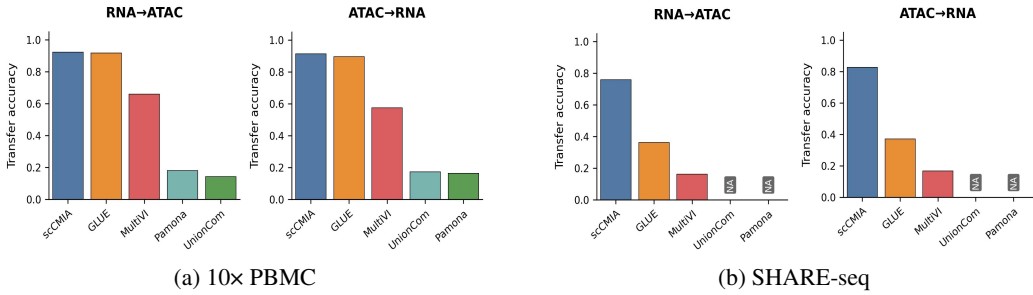

(a) 10× PBMC             (b) SHARE-seq

Figure 2: Bidirectional label transfer accuracy across integration methods.

a kNN classifier to transfer cell type labels from scRNA-seq to scATAC-seq data (RNA-to-ATAC transfer) and vice versa (ATAC-to-RNA transfer). The comparative results, as summarized in Fig. 2, demonstrate that our scCMIA method significantly outperformed existing methods in both transfer directions. Notably, the high concordance observed between transferred labels across modalities provides strong evidence for successful cross-modal alignment.

## 4.3 RECONSTRUCTION EXPERIMENTS

For the data reconstruction evaluation, we performed systematic experiments to validate scCMIA's capability to accurately reconstruct both the scRNA-seq and scATAC-seq data spaces. As evidenced in Tables 2 and 3, scCMIA demonstrates superior multimodal reconstruction performance compared to benchmark methods, achieving the lowest error metrics across both scRNA and scATAC modalities.

Table 2: Reconstruction performance (mean ± std) on scRNA across four random seeds. Lower is better (↓). **Bold** = best, *italic* = second best.

| Method | Align | Recon | Metric | SHARE-seq | SNARE-seq | 10×PBMC | ISSAAC-seq |
|--------|-------|-------|--------|-----------|-----------|---------|------------|
| GLUE | ✓ | ✓ | RMSE | 0.5214±0.013 | 0.8298±0.278 | 1.8103±0.005 | 1.3244±0.001 |
|  |  |  | MAE | 0.0978±0.001 | 0.0272±0.136 | **0.1494±0.001** | 0.4807±0.001 |
| MultiVI | ✓ | ✓ | RMSE | 0.9385±0.001 | 0.9322±0.003 | **0.9508±0.001** | 3.5434±0.154 |
|  |  |  | MAE | 0.3018±0.02 | 0.2665±0.005 | 0.3061±0.004 | 2.4638±0.111 |
| Cobolt |  | ✓ | RMSE | *0.3832±0.021* | 0.8319±0.003 | 1.8127±0.001 | 1.3261±0.001 |
|  |  |  | MAE | **0.0300±0.001** | **0.0268±0.005** | *0.1499±0.013* | 0.4814±0.001 |
| scMM |  | ✓ | RMSE | 0.4824±0.003 | *0.6309±0.008* | 1.7939±0.014 | 1.2817±0.035 |
|  |  |  | MAE | 0.0905±0.0003 | 0.1228±0.0003 | 0.2945±0.007 | 0.2525±0.04 |
| scButterfly |  | ✓ | RMSE | 2.1405±0.048 | 1.3840±0.033 | 3.7336±0.050 | *1.2452±0.027* |
|  |  |  | MAE | 0.1464±0.041 | *0.0750±0.001* | 0.2756±0.024 | *0.1918±0.012* |
| **scCMIA** | ✓ | ✓ | RMSE | **0.3213±0.025** | **0.5490±0.003** | *1.0140±0.013* | **0.8931±0.001** |
|  |  |  | MAE | *0.0624±0.014* | 0.0919±0.046 | 0.1591±0.080 | **0.0750±0.043** |

Table 3: Reconstruction performance (mean ± std) on scATAC across four random seeds. Lower is better (↓). **Bold** = best, *italic* = second best.

| Method | Align | Recon | Metric | SHARE-seq | SNARE-seq | 10×PBMC | ISSAAC-seq |
|---|---|---|---|---|---|---|---|
| MultiVI | ✓ | ✓ | RMSE | 0.3148±0.001 | 0.2784±0.003 | *1.4572±0.0002* | 1.7726±0.005 |
| | | | MAE | 0.1655±0.001 | 0.2045±0.010 | 0.9447±0.004 | 0.4512±0.004 |
| Cobolt | | ✓ | RMSE | *0.2621±0.001* | *0.2460±0.001* | 1.5283±0.002 | 2.1371±0.161 |
| | | | MAE | **0.0406±0.0003** | 0.0623±0.001 | 0.7039±0.013 | 0.7646±0.005 |
| scMM | | ✓ | RMSE | 0.3730±0.0002 | 0.3451±0.0004 | 1.6689±0.0002 | 3.0510±0.103 |
| | | | MAE | 0.1202±0.0001 | 0.0998±0.0002 | 0.8064±0.002 | *0.4316±0.0001* |
| scButterfly | | ✓ | RMSE | 0.4942±0.001 | 0.5001±0.022 | 1.4746±0.016 | *1.6950±0.008* |
| | | | MAE | 0.4924±0.024 | 0.4995±0.131 | **0.4688±0.021** | 0.4931±0.011 |
| **scCMIA** | ✓ | ✓ | RMSE | **0.2607±0.003** | **0.2459±0.001** | **1.2086±0.016** | **1.1996±0.0001** |
| | | | MAE | *0.0532±0.145* | **0.0561±0.123** | *0.6713±0.019* | **0.2404±0.0167** |

Notably, while scCMIA maintained top-2 ranking on the 10× PBMC dataset reconstruction task, it exhibited marginally higher error metrics compared to GLUE and MultiVI. The observed performance differences were quantitatively minimal, with MAE and RMSE discrepancies of merely 0.0097 and 0.0632 respectively against GLUE. This narrow performance gap suggests comparable reconstruction fidelity among the top-performing methods, while scCMIA maintains a distinct advantage in its capability for simultaneous multimodal reconstruction. Furthermore, the comprehensive cross-modal reconstruction performance across all evaluated datasets confirms scCMIA's methodological strength in preserving data integrity during integration processes.

### 4.4 MODEL VALIDITY EXPERIMENTS

**Decoupling Effectiveness** The effectiveness of our model's decoupling mechanism is validated through two quantitative approaches. Our primary validation directly assesses the objective of our training strategy by measuring the MI between the resulting latent variables. This confirms that MI was successfully minimized between semantic and modality-specific representations within a modality, while being maximized between the semantic representations across modalities. As an auxiliary method, we also calculate the cosine similarity between these vectors to further verify their statistical independence.

The experimental results are shown in Table 4, which shows that the modal specificity and semantic MI in the single mode are close to 0, which also means that the more independent the two random variables are. The semantic mutual information of the two modalities after RtA has increased by 789. 19% significantly compared to the mutual information within a single modal. Furthermore, the cosine similarity calculated between modality-specific and semantic representations is also nearly zero. The decoupled semantic representation has a higher cosine similarity, which also indicates the consistency of the two representations in terms of direction. These findings strongly validate that the disentangled variables are independent and uncorrelated, which confirms the effectiveness of our decoupling approach. It is also consistent with the assumptions of Eq. 4.

Additionally, we further investigated decoupled modality-specific and semantic representations using clustering tasks. The experimental results are shown in Fig. 3. The results suggest that the semantic representations contain more information related to cell identity, whereas the modality-specific portion of the representations contain less or almost no (vs. scATAC) information related to cell identity. In particular, both the semantic representations of scRNA and scATAC are rich in cell identity information. Since modality-specific data include unique information pertinent to each modality, this information is essential for subsequent reconstruction tasks to achieve better performance.

**Ablation Study** We also conducted ablation studies on each module of the scCMIA by constructing models that include different combinations of modules. Specifically, the CLUB module is used for intra-modal decoupling, while the RtA and CrossVQ modules are employed for modal alignment and to address the issue of insufficient information within individual modalities. We constructed ablation experiments with different components (in the SHARE-seq dataset) and the experimental results

Table 4: Comparison of dependency and direction consistency metrics across multi-omics datasets. Higher values indicate stronger semantic alignment.

| Metric | | Dataset | | | |
|---|---|---|---|---|---|
| Category | Type | SHARE-seq | SNARE-seq | 10× PBMC | ISSAAC-seq |
| Dependency (MI) | Sem-Sp (scRNA) | 0.0181 | 0.0127 | 0.0791 | 0.0484 |
| | Sem-Sp (scATAC) | 0.0030 | 0.0037 | 0.0048 | 0.0113 |
| | Sem-Sem (RtA) | **0.2466** | **0.1761** | **0.2751** | **0.2226** |
| Direction Consistency (Cosine Sim.) | Sem-Sp (scRNA) | 0.0264 | 0.0134 | -0.0097 | -0.0044 |
| | Sem-Sp (scATAC) | 0.0026 | 0.00001 | 0.0090 | 0.0007 |
| | Sem-Sem (Raw) | -0.0049 | -0.0017 | -0.0039 | -0.0052 |
| | Sem-Sem (RtA) | **0.6087** | **0.5070** | **0.6013** | **0.6818** |

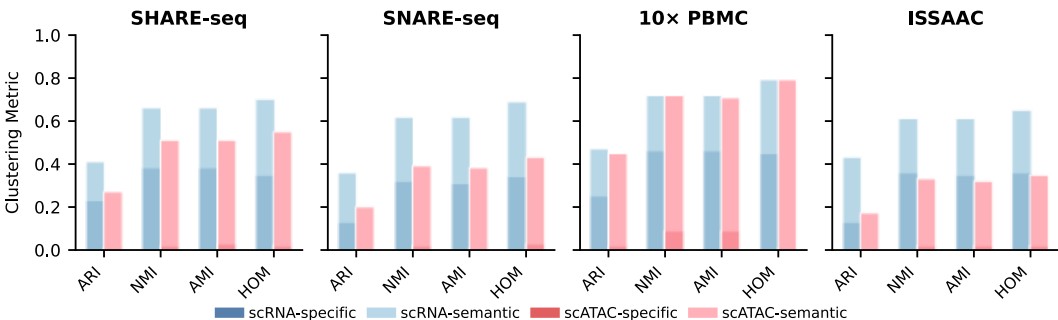

Figure 3: Comparison of clustering results of decoupled **modality-specific** and **semantic** representations across different datasets.

are shown in Table 5. The experimental results were able to observe that the inclusion of the RtA module significantly improves cross-modal alignment, while the inclusion of CrossVQ improves the performance in terms of reconstruction. In addition, the model containing all components (CrossVQ, CLUB, RtA) is able to achieve good performance in both alignment and reconstruction performance.

In addition, we also compare the performance of CLUB+RtA and Full model on the label migration and clustering tasks, and the experimental results are shown in Fig. 4, which shows that scCMIA with Full model is able to have better performance, and the experiments also validate the reasonableness of the individual modules that we have designed. Together, these modules provide a robust framework for handling multi-modal data integration and analysis.

Table 5: Ablation experiments of different modules, among which scCMIA is a full model, including CLUB+RtA+CrossVQ. (FOSCTTM↓, RMSE↓, MAE↓)

| Components | FOSCTTM | scRNA | | scATAC | |
|---|---|---|---|---|---|
| | | RMSE | MAE | RMSE | MAE |
| VQ-VAE (Baseline) | 0.4801 | 0.4557 | 0.1060 | 0.2634 | 0.1195 |
| + CrossVQ, CLUB | 0.4945 | **0.3125** | 0.0629 | 0.2677 | 0.0544 |
| + CrossVQ, RtA | 0.0231 | 0.3666 | **0.0612** | 0.2625 | 0.0539 |
| + CLUB, RtA | 0.0178 | 0.3234 | 0.0641 | 0.2612 | 0.0592 |
| scCMIA | **0.0132** | *0.3213* | *0.0624* | **0.2607** | **0.0532** |

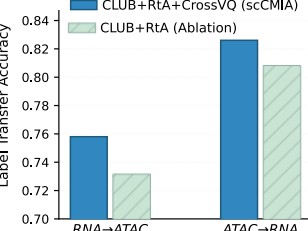

Figure 4: Compare the performance of scCMIA and CLUB + RtA modules on the label transfer task.

## 4.5 DOWNSTREAM FIDELITY ASSESSMENT

To quantitatively evaluate integration quality through downstream label-transfer fidelity, we adopt the cell-type classification task as defined in Liu et al. Liu et al. (2025), where latent representations

from multimodal integration are used to transfer annotations across batches. Specifically, for each dataset and modality configuration (RNA and ATAC, and bimodal RNA+ADT), we perform 5-fold cross-validation: in each fold, one batch (or subset when only one batch exists) serves as the query, while the remaining data constitute the reference for training. A standard multilayer perceptron (MLP) classifier is trained on the reference embedding and applied to predict cell-type labels in the query. Finally, the methods are ranked using the comprehensive scores (**Rank**).

On the 10× BMMC datasets, scCMIA delivers the strongest overall performance, achieving the best scores on every metric and ranking first among baselines including scBridge Li et al. (2023), scMoGNN Wen et al. (2022), scMM, and MultiVI (Tables 6, 7).

Table 6: Cell-type classification performance on RNA, evaluated via 5-fold MLP-based classification task(best in **bold**).

| Method | OAC | AAC | Spec. | F1 | Rank |
|---|---|---|---|---|---|
| scCMIA | 0.8900 ± 0.0031 | **0.8585 ± 0.0117** | 0.9945 ± 0.0002 | **0.8611 ± 0.0073** | **1** |
| scMM | **0.8907 ± 0.0035** | 0.8399 ± 0.0125 | **0.9946 ± 0.0002** | 0.8449 ± 0.0084 | 2 |
| MultiVI | 0.8609 ± 0.0033 | 0.7966 ± 0.0104 | 0.9931 ± 0.0002 | 0.8042 ± 0.0067 | 3 |
| scMoGnn | 0.8030 ± 0.0019 | 0.7163 ± 0.0077 | 0.9901 ± 0.0001 | 0.7221 ± 0.0102 | 4 |
| scBridge | 0.7520 ± 0.0049 | 0.6569 ± 0.0030 | 0.9876 ± 0.0001 | 0.6690 ± 0.0065 | 5 |
| scButterfly | 0.6938 ± 0.0052 | 0.6021 ± 0.0178 | 0.9846 ± 0.0003 | 0.6160 ± 0.0151 | 6 |

Table 7: Cell-type classification performance on ATAC, evaluated via 5-fold MLP-based classification task (best in **bold**).

| Method | OAC | AAC | Spec. | F1 | Rank |
|---|---|---|---|---|---|
| scCMIA | **0.8923 ± 0.0033** | **0.8459 ± 0.0083** | **0.9947 ± 0.0001** | **0.8509 ± 0.0072** | **1** |
| scMM | 0.8821 ± 0.0024 | 0.8230 ± 0.0111 | 0.9942 ± 0.0001 | 0.8311 ± 0.0082 | 2 |
| MultiVI | 0.8705 ± 0.0026 | 0.7982 ± 0.0076 | 0.9936 ± 0.0001 | 0.8099 ± 0.0063 | 3 |
| scBridge | 0.8488 ± 0.0022 | 0.7724 ± 0.0046 | 0.9925 ± 0.0001 | 0.7854 ± 0.0040 | 4 |
| scMoGnn | 0.4879 ± 0.0049 | 0.2525 ± 0.0071 | 0.9734 ± 0.0002 | 0.2290 ± 0.0099 | 5 |
| scButterfly | 0.2954 ± 0.0059 | 0.1268 ± 0.0027 | 0.9633 ± 0.0003 | 0.1152 ± 0.0035 | 6 |

For CITE-seq (RNA+ADT), Table 8 compares scCMIA against TotalVI Gayoso et al. (2021) and sciPENN Lakkis et al. (2022), Under the same 5-fold MLP classification protocol, scCMIA again attains the highest scores in all metrics, notably improving Average Accuracy (AAC) by +2.82% and F1 by +3.17% over TotalVI—the strongest baseline—highlighting its efficacy in multimodal joint embedding and discriminative feature learning. The consistent lead across unimodal and multimodal settings confirms scCMIA's versatility and integration quality.

Table 8: Cell-type classification performance on RNA+ADT (CITE-seq), evaluated via 5-fold MLP-based classification task (best in **bold**).

| Method | OAC | AAC | Spec. | F1 | Rank |
|---|---|---|---|---|---|
| scCMIA | **0.8666 ± 0.0036** | **0.7373 ± 0.0063** | **0.9969 ± 0.0001** | **0.7429 ± 0.0066** | **1** |
| TotalVI | 0.8608 ± 0.0025 | 0.7171 ± 0.0102 | 0.9968 ± 0.0001 | 0.7201 ± 0.0077 | 2 |
| sciPENN | 0.8344 ± 0.0019 | 0.6775 ± 0.0132 | 0.9962 ± 0.0001 | 0.6676 ± 0.0153 | 3 |

## 5 CONCLUSION

This paper introduces scCMIA, a novel self-supervised framework designed to address the challenges of integrating single-cell multi-omics data. Based on mutual information principles and a unified discrete codebook, this model not only outperforms existing methods in alignment and reconstruction tasks but also pioneers the use of its interpretable latent space for biological exploration. It successfully quantifies regulatory conservation and coupling differences across distinct cell lineages, demonstrating its immense potential as a tool for biological discovery.

**Ethics statement**. This work adheres to the ICLR Code of Ethics. Our research is based on publicly available, anonymized datasets and we foresee no direct negative societal impacts or ethical concerns.

**Reproducibility statement**. All code, model architecture details, and data preprocessing steps required to reproduce our findings are provided in the supplementary materials and detailed in the Appendix.

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

## A   TECHNICAL APPENDICES AND SUPPLEMENTARY MATERIAL

This appendix provides a comprehensive overview of the technical implementation of scCMIA. This section includes detailed diagrams of the model architecture, a complete list of training hyperparameters, a description of the data preprocessing pipeline, and the algorithm's pseudocode to ensure full reproducibility.

### A.1   MORE RELATED WORKS

**Maximize Mutual Information Lower Bound**. MI The mutual information neural estimator (MINE) Belghazi et al. (2018b) relies on kernel density estimation of random variables and estimates MI through a neural network to fit the expectation of two distributions. Deep InfoMax (DIM) Hjelm et al. (2019) utilizes autoencoders to learn latent representations of two variables, then estimates MI by maximizing the consistency between the representations. InfoNCE Oord et al. (2018a) estimates a lower bound on MI by contrasting the similarity of representations between positive and negative samples. InfoNCE can be combined with deep neural networks to learn complex representations and estimate highly nonlinear MI relationships.

**Minimize Mutual Information Upper Bound**. MI minimization has found wide applications in the disentangled representation learning Von Kügelgen et al. (2021), domain adaptation Vettoruzzo et al. (2024), and information bottleneck methods Tian et al. (2020). However, these methods require closed-form density functions and tractable log-density ratios between the joint and marginal distributions, which limits the exact computation of MI to a few special cases. To address this challenge, sample-based MI estimators Belghazi et al. (2018a); Cherti et al. (2023) have been proposed. For example, L1out Poole et al. (2019a) can provide more accurate MI estimates with large sample sizes. However, when applied to MI minimization models, it suffers from high numerical instability. The contrastive log-ratio upper bound (CLUB) Cheng et al. (2020) is a reliable MI estimator that can also be trained within a gradient descent framework. To obtain a tighter upper bound on the MI Yang et al. (2024), we use CLUB to evaluate the upper boundary for better intra-modal decoupling.

## A.2 scCMIA Algorithm

Algorithm 1 provides a clear step-by-step outline of the scCMIA algorithm, emphasizing the key steps such as intra-modal decoupling, cross-modal alignment using contrastive learning and VQ operations, and the calculation and minimization of the total loss term $\mathcal{L}_{\text{scCMIA}}$ to optimize the model parameters $\theta$.

---

**Algorithm 1** scCMIA

---

1: **Input:** scRNA $X$, scATAC $Y$, epoch $N$,
2:    scRNA modality-specific encoder $\Phi$ and semantic encoder $\hat{\Phi}$,
3:    scATAC modality-specific encoder $\Psi$ and semantic encoder $\hat{\Psi}$,
4:    model parameter $\theta$.
5: Initialize codebook $e \in \{e_1, e_2, \ldots, e_k\}$, learnable parameters $W$
6: **for** $i = 1$ **to** $N$ **do**
7:    $\hat{I}_{\text{vCLUB}}, x, \hat{x}, y, \hat{y} \leftarrow \text{CLUB}(X, Y, \Phi, \hat{\Phi}, \Psi, \hat{\Psi})$
8:    $\mathcal{L}_{\text{RtA}}, x', y' \leftarrow \text{RtA}(x, y)$
9:    $\mathcal{L}_{\text{CrossVQ}}, x'', y'' \leftarrow \text{CrossVQ}(x', y')$
10:    $\mathcal{L}_{\text{rec}} \leftarrow \text{Reconstruct\_decoder}(x'', \hat{x}, y'', \hat{y})$
11: **end for**
12: $\mathcal{L}_{\text{scCMIA}} \leftarrow \hat{I}_{\text{vCLUB}} + \mathcal{L}_{\text{RtA}} + \mathcal{L}_{\text{CrossVQ}} + \mathcal{L}_{\text{rec}}$
13: $\theta \leftarrow \arg\min_\theta \mathcal{L}_{\text{scCMIA}}$

---

## A.3 More Experimental Supplements

## A.4 Datasets

Each dataset contains variable sample sizes with different number of cell-types and the dimensions of both scRNA and scATAC are high. In Table 9, we present detailed information about the datasets used in this work, including sample size, dimensions of paired modalities, and cell types.

Table 9: Composition of the experimental datasets.

| Dataset | Cells (n) | Genes | Peaks | Proteins | Cell Types |
|---|---|---|---|---|---|
| 10x PBMC | 9,631 | 29,095 | 107,194 | — | 19 |
| SHARE-seq | 32,231 | 21,478 | 340,341 | — | 22 |
| SNARE-seq | 9,190 | 28,930 | 241,757 | — | 22 |
| ISSAAC-seq | 10,361 | 32,285 | 169,180 | — | 23 |
| 10x BMMC | 69,249 | 13,431 | 116,490 | — | 22 |
| BMMC (CITE-seq) | 90,261 | 13,953 | — | 134 | 45 |

Given the high dimensionality and sparsity issues prevalent in both scRNA and scATAC data, it is necessary to perform feature selection beforehand to better handle the data. For scRNA data, we select 2000 highly variable genes, while for scATAC data, we choose 30,000 high-variance regions as features.

## A.5 Evaluation Metrics

### A.5.1 Performance Evaluation Metrics

This section introduces the calculation formulas for the Fraction of Samples Closer Than the True Match (FOSCTTM) and matching accuracy (MA).

FOSCTTM is a core alignment performance metric. It is specifically designed to evaluate whether data from two different modalities at the single-cell level has been accurately matched together. A lower FOSCTTM value indicates higher alignment precision of the model. If $N$ cells have true pairwise information, FOSCTTM is defined as

$$\text{FOSCTTM} = \frac{1}{2N}\left(\sum_{i=1}^{N}\frac{n_1^{(i)}}{N} + \sum_{i=1}^{N}\frac{n_2^{(i)}}{N}\right),$$
$$n_1^{(i)} = |\{j \mid d(\mathbf{x}_j, \mathbf{y}_i) < d(\mathbf{x}_i, \mathbf{y}_i)\}|,$$
$$n_2^{(i)} = |\{j \mid d(\mathbf{x}_i, \mathbf{y}_j) < d(\mathbf{x}_i, \mathbf{y}_i)\}|. \tag{13}$$

The parameters in the formula are explained as follows:

- $d(\cdot, \cdot)$: A function used to calculate the Euclidean distance.

- $n_1^{(i)}$ and $n_2^{(i)}$: Denote the number of cells that are closer to the $i$-th sample than their true match in the opposite dataset.

- The value of FOSCTTM is in the range of $[0, 1]$. Smaller values of FOSCTTM indicate better performance.

Additionally, we tested the accuracy of correctly matching paired samples of another modality under given batch samples from different modal perspectives. We use the MA as a measure, which is defined by the formula:

$$\mathbf{MA} = \frac{1}{N}\frac{1}{B}\sum_{k=1}^{N}\sum_{i=1}^{B}\sum_{j=1}^{B}\mathbb{I}\left(\sum_{m\in\{\mathbf{x},\mathbf{y}\}}\text{Cos}_{\text{sim}}(m_i, m_i) > \sum_{m\in\{\mathbf{x},\mathbf{y}\}}\text{Cos}_{\text{sim}}(m_j, m_i) \text{ for all } j \neq i\right). \tag{14}$$

The parameters in the formula are explained as follows:

- $N$: Total number of samples in the dataset.

- $B$: Batch size (number of samples in each batch).

- $\mathbf{x}_i$ and $\mathbf{y}_i$: Represent the two modalities (e.g., scRNA and scATAC) of the $i$-th sample.

- $\text{Cos}_{\text{sim}}(\cdot, \cdot)$: Cosine similarity function.

- $\mathbb{I}(\cdot)$: Indicator function, which takes the value 1 if the condition is true, and 0 otherwise.

- Range of values: $[0, 1]$.

### A.5.2 BIOLOGICAL INTERPRETABILITY METRICS

CTSI (Cell Type Specificity Index) and Conservation Score are used to reveal whether components such as the model's latent space and encoding capture meaningful patterns consistent with biological knowledge.

Conservation Score measures the frequency with which both scRNA and scATAC modalities map to the same discrete code within the same cell. It reveals the degree of coupling in multimodal regulation within different cell types. Variations in its values constitute biological discoveries, demonstrating that the model has learned deep insights into the distinct regulatory logic of different cell types. It combines two parts: the overlap of codes used and the similarity of their usage frequency distributions. A higher score indicates that two cell types utilize the VQ codebook in a more similar or conserved manner.

Overlap Score Measures the similarity between the sets of VQ codes used by two cell types, $C_i$ and $C_j$.

$$\text{Overlap}(C_i, C_j) = \frac{|C_i \cap C_j|}{|C_i \cup C_j|}. \tag{15}$$

Distribution similarity measures the similarity between the VQ code frequency vectors, $F_i$ and $F_j$, for two cell types.

$$\text{DistSim}(F_i, F_j) = \frac{F_i \cdot F_j}{\|F_i\|\|F_j\|} = \frac{\sum_{k=1}^{N} F_{ik}F_{jk}}{\sqrt{\sum_{k=1}^{N} F_{ik}^2}\sqrt{\sum_{k=1}^{N} F_{jk}^2}}. \tag{16}$$

Conservation Score is a weighted average of the Overlap Score and the Distribution Similarity. The $\alpha$ is a weighting factor, we set $\alpha = \frac{1}{2}$.

$$\text{Conservation}(i, j) = \alpha \cdot \text{Overlap}(C_i, C_j) + (1 - \alpha) \cdot \text{DistSim}(F_i, F_j). \tag{17}$$

Additionally, CIST measures the specificity with which different cell types utilize specific codes within a unified codebook. A high CTSI value indicates the codebook has successfully learned discrete states that distinguish distinct cellular identities, providing a biological interpretation for the model's black-box interior. It aims to quantify how specific the VQ code usage is for a particular cell type compared to all other cell types. A high CTSI score for a cell type suggests it uses at least one VQ code with a much higher frequency than any other cell type does, indicating a specific signature in the codebook space.

Let $F_i$ be the frequency vector of VQ codes for cell type $i$, and $\mathcal{T}$ be the set of all cell types. The CTSI for cell type $i$ is defined as:

$$\text{CTSI}_i = \frac{\max(F_i) - \max\left(\frac{1}{|\mathcal{T}|-1}\sum_{j \in \mathcal{T}, j \neq i} F_j\right)}{\max(F_i)}. \tag{18}$$

The parameters in the formula are explained as follows:

- $\max(F_i)$ is the maximum frequency of any single code for cell type $i$.
- $\frac{1}{|\mathcal{T}|-1}\sum_{j \in \mathcal{T}, j \neq i} F_j$ is the average frequency vector across all other cell types.
- $\max(\cdot)$ of that average vector gives the highest frequency achieved for any code on average by other cell types.

A.6 MODEL ARCHITECTURE AND IMPLEMENTATION DETAILS

In order to improve the reproducibility of the algorithm for easy understanding, we show the architecture of the modal model and the parameter settings in detail in Fig. 5. This includes the modality-specific and semantic coders, the RtA module, and the decoders for each modality.

We implemented scCMIA on an NVIDIA RTX A6000. First, to reduce the model parameters and remove redundant information, we preprocessed scRNA (Principal Component Analysis, PCA) and scATAC (Latent Semantic Indexing, LSI) using a linear-dimensionality reduction method. The raw scRNA and scATAC data are reduced to 256 dimensions and used as input for the scCMIA model. We use Adam as the optimizer with the learning rate set to 0.00001. In the training phase, 10% of the cells were used as a validation set, the number of training iterations was set to 500, we implemented an early stopping mechanism that halts training if the loss does not decrease for 20 consecutive epochs. In addition, to validate the robustness of our method, we set up four different random seeds for the experiments.

The experimental design accounted for methodological differences among comparative frameworks: 1) GLUE exclusively relies on RNA-derived association graphs for cross-modal integration, inherently limiting its scATAC reconstruction capacity (RNA reconstruction metrics only reported); 2) The original Cobolt implementation lacked intrinsic reconstruction functionality, necessitating our implementation of a dedicated reconstruction module to enable fair performance comparison.

To assess the preservation of local structural correspondence between RNA and ATAC modalities, we conducted bidirectional label transfer experiments (RNA→ATAC and ATAC→RNA) following established protocols from reference methods (e.g., Pamona, UnionCom). A k-nearest neighbors (KNN) classifier was trained on low-dimensional embeddings of the source modality to ensure consistency with benchmark implementations. The trained model was then applied to predict cell-type labels using the target modality's embeddings, thereby enabling cross-modality prediction. Label

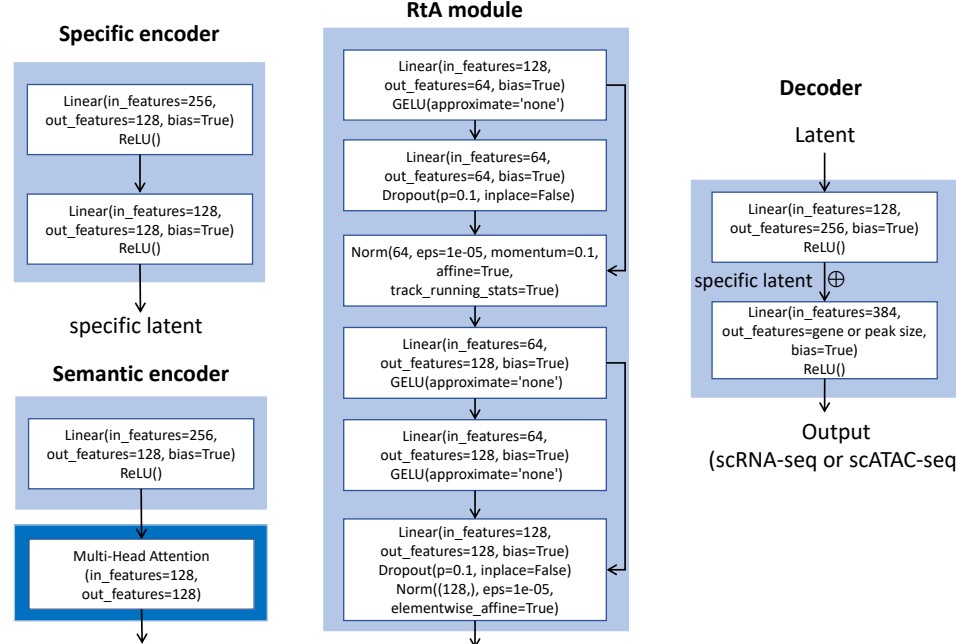

Figure 5: Model structure details. The model architectures and parameter configurations of the domain-specific encoder, semantic encoder, RtA module, and decoder in our proposed framework.

transfer accuracy was systematically quantified across all annotated cell types to evaluate alignment fidelity between modalities. This rigorous framework not only facilitates direct comparison with prior studies but also objectively measures the mutual translatability of cellular state representations across distinct data types, providing critical insights into multimodal integration performance.

**Classifier Task.** To ensure clarity and reproducibility, we strictly separate the methodology into two distinct stages: (A) the internal, unsupervised training of our integration model, and (B) the external, supervised evaluation of the resulting embeddings via downstream tasks.

- **(A) Model Training and Hyperparameter Selection.** The primary goal of this stage is to learn a latent space representation (embedding) that optimally reconstructs the original omics data, without using any cell type labels. Crucially, no cell type annotations were used during this stage, preventing any label information leakage. Once the optimal hyperparameters were determined and the model was trained, its parameters were frozen. This fixed, pre-trained model was then used to generate embeddings for all subsequent evaluation tasks.

- **(B) Evaluation of cell type classification.** To assess the quality of the frozen embeddings produced by our method (and all baseline methods), we performed a supervised cell type classification task. This task evaluates how well each embedding preserves biological identity (i.e., cell type) while successfully correcting for technical batch effects. We adopted a rigorous cross-validation strategy based on the data batches, inspired by the protocol established by Liu et al. Liu et al. (2025). This evaluation was performed after the integration, using the generated low-dimensional embeddings as input features for the classifier. The final classification performance was calculated by averaging the evaluation metrics (e.g., OAC, ACA, F1-Score) across all N folds. This process robustly measures the ability of each integration method to produce a batch-invariant and biologically meaningful latent space.

## B  SUPPLEMENTARY EXPERIMENTAL RESULTS

This includes a quantitative analysis of how data reconstruction improves data quality, a masking experiment to verify the model's imputation accuracy, a deeper exploration of the biological insights enabled by our model's interpretable discrete codebook, and a visualization of the total loss curves to demonstrate stable model convergence across all datasets.

## B.1 Reconstruction tasks can effectively enhance data quality

To evaluate whether the reconstruction module of scCMIA can effectively improve the quality of raw single-cell data, we compared key data metrics before and after model reconstruction, with particular focus on the issue of data sparsity.

Table 10: Key metrics of scRNA and scATAC data were compared before and after scCMIA reconstruction.

| Data Type | Metric | Raw | Reconstructed | Improvement (%) |
|---|---|---|---|---|
| scRNA | Density (%) | 9.3123 | 9.4842 | +1.8 |
| | Avg. expressed genes/cell | 186 | 189 | +1.7 |
| | Avg. expressed cells/gene | 896 | 913 | +1.9 |
| scATAC | Density (%) | 20.2984 | 38.3894 | +89.1 |
| | Avg. expressed peaks/cell | 6089 | 11516 | +89.1 |
| | Avg. expressed cells/peak | 1954 | 3697.3 | +89.2 |

As shown in the Table. 10, after reconstruction by scCMIA, the data quality of both scRNA-seq and scATAC-seq modalities was significantly improved. The experimental results clearly demonstrate that the reconstruction process effectively reduces data sparsity. For scRNA-seq data, all metrics showed modest yet robust improvements. For scATAC-seq data, which suffers from more severe sparsity, both data density and feature detection rates increased by approximately 89%, demonstrating particularly significant effects. This confirms that our reconstruction module can generate a more complete and information-rich cellular landscape by filling in technologically lost information.

Although the above experiments demonstrate that reconstruction can increase data density, we must verify that this improvement stems from accurate data imputation rather than the filling of random noise. To this end, we designed a masking experiment to directly evaluate the model's imputation capability. We conducted simulations on the 10x PBMC dataset. For each cell, we randomly masked 10% to 30% of its feature values. To ensure fairness in evaluation, masked positions were strictly balanced: half were original non-zero values (to test false negatives/recall), and half were original zero values (to test false positives). The model's task is to predict these masked values. We framed this as a binary classification problem and used Recall and AUROC as evaluation metrics.

Table 11: Interpolation performance under varying masking ratios.

| Masking Ratio | scRNA Recall | scRNA AUROC | scATAC Recall | scATAC AUROC |
|---|---|---|---|---|
| 10% | 0.7361 | 0.8344 | 0.9643 | 0.8119 |
| 15% | 0.7325 | 0.8333 | 0.9643 | 0.8114 |
| 20% | 0.7330 | 0.8323 | 0.9642 | 0.8108 |
| 25% | 0.7261 | 0.8289 | 0.9641 | 0.8106 |
| 30% | 0.7244 | 0.8283 | 0.9640 | 0.8103 |

As shown in the Table. 11 , scCMIA demonstrates robust and powerful interpolation performance even under varying masking ratios. This experiment provides direct quantitative evidence that the increased data density observed earlier is not an artifact but rather a reflection of the model's strong and precise interpolation capabilities. Particularly on scATAC-seq data, the model correctly recovered approximately 96% of genuinely open chromatin regions (peaks) that were artificially obscured. This robustly confirms that our reconstruction process provides an enhanced, biologically more accurate representation of cellular states by recovering true signals from technical noise.

### B.1.1 Biological Significance Validation and Application Exploration of Unified Codebooks

To validate the biological significance and practical value of the VQ-VAE framework and unified codebook in this study, we designed two supplementary experiments. The first experiment aimed to verify whether the discrete representations learned by the codebook itself possess interpretable biological structures. The second experiment further explored the potential of leveraging these structures for downstream biological knowledge discovery.

First, we quantitatively assessed the intrinsic properties of the unified codebook by introducing two novel metrics: CTSI and Consistency Rate. Experimental results (Fig. 12) demonstrate that high CTSI values (most $> 0.8$) confirm the codebook learned highly specialized, non-generalized discrete encodings for different cell types. Simultaneously, the substantial variation in Consistency Rate across cell types (e.g., as high as 0.98 in memory B cells versus as low as 0.33 in plasma cells) reveals the model's ability to successfully capture diverse regulatory coupling relationships between transcriptomes and chromatin accessibility across cell types, effectively avoiding excessive or forced alignment across different biological states Liggett & Sankaran (2020); Chi et al. (2024). This analysis fundamentally validates the unified codebook as a structured, interpretable layer of biological representation.

Table 12: Cell-type-specific integration (CTSI) scores and consistency rates across modalities.

| Cell Type | scRNA CTSI | scATAC CTSI | Consistency Rate |
|---|---|---|---|
| CD14 Mono | 0.7299 | 0.6116 | 0.3751 |
| CD4 Naive | 0.8448 | 0.8133 | 0.8603 |
| CD8 Naive | 0.8569 | 0.8365 | 0.0185 |
| CD8 TEM_1 | 0.7848 | 0.7578 | 0.6429 |
| HSPC | 0.8127 | 0.8473 | 0.6471 |
| Intermediate B | 0.8664 | 0.8684 | 0.8500 |
| Memory B | 0.8685 | 0.8449 | 0.9765 |
| NK | 0.8577 | 0.8378 | 0.9280 |
| Naive B | 0.8747 | 0.8716 | 0.8560 |
| Plasma | 0.7931 | 0.8632 | 0.3333 |

Second, building upon the validated unified codebook, we introduced the regulatory Conservation Score (CS) to quantify the similarity of regulatory programs across cell types, aiming to test the model's capability for biological knowledge discovery. This results ((Fig. 13)) successfully reproduced known cellular lineage relationships, cells within the same lineage (e.g., B cell subpopulations) obtained high RCS scores, while scores between different lineages (e.g., lymphoid and myeloid) were lower. More importantly, this approach reveals finer biological insights, such as quantitatively distinguishing functional differences among distinct monocyte subpopulations and capturing shared cytotoxic programs between NK cells and CD8 TEM_1 cells. This experiment demonstrates that our model transcends mere data integration, serving as a quantitative exploration tool to generate novel insights into cellular regulatory networks.

Table 13: Conservation scores (CS) and biological interpretation across cell type pairs. Higher CS indicates stronger cross-modality alignment.

| Category | Cell Type Pair | RNA CS | ATAC CS | Interpretation |
|---|---|---|---|---|
| High Conservation (B cell) | Naive B vs. Plasma | 0.774 | 0.847 | Strong conservation in B-cell development |
| High Conservation (T cell) | CD8 Naive vs. CD8 TEM_1 | 0.510 | 0.551 | Conservation across T-cell subtypes |
| Low Conservation (Distant) | CD4 Naive vs. CD14 Mono | 0.180 | 0.095 | Lymphoid vs. myeloid programs |
| Nuanced Insights | CD14 Mono vs. CD16 Mono | 0.361 | 0.365 | Subtle differences between monocyte subtypes |
| | NK vs. CD8 TEM_1 | 0.410 | 0.509 | Shared cytotoxic program |

To sum up, these two complementary experiments form a complete chain of reasoning. The first experiment establishes the structural validity and interpretability of the discrete code book in our method, demonstrating it is not a complex component designed for novelty's sake. The second experiment demonstrates the functional utility of this structure, proving it can serve as a powerful tool for discovering and quantifying cellular regulatory logic. Together, they confirm that our proposed scCMIA framework not only excels at alignment and reconstruction tasks but also delivers profound biological insights, opening new analytical dimensions for single-cell multi-omics research.

### B.1.2 MODEL CONVERGENCE ANALYSIS

The overall training objective function of the cCMIA framework comprises multiple components, resulting in a complex training process. To validate the model's convergence, we visualized the evolution of the total training loss across epochs on all four benchmark datasets (10x Multiome PBMC, SHARE-seq, SNARE-seq, ISSAAC-seq).

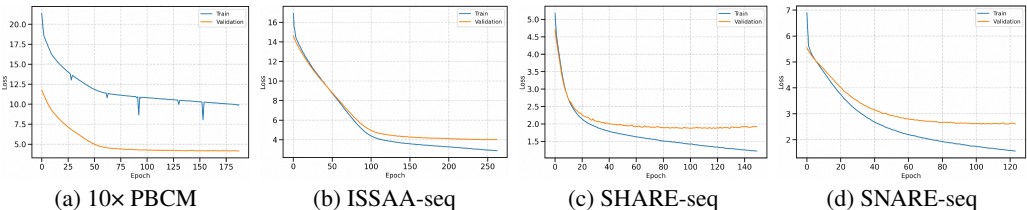

(a) 10× PBCM     (b) ISSAA-seq     (c) SHARE-seq     (d) SNARE-seq

Figure 6: Total loss ($\mathcal{L}_{scCMIA}$) convergence curve of scCMIA across four datasets.

As shown in Fig. 6, the model's training process remains stable even across datasets with varying complexity and feature differences. This provides robust assurance for scCMIA's broad applicability across diverse single-cell multi-omics datasets.

### B.1.3 ALIGNMENT PERFORMANCE EXPERIMENT SUPPLEMENT

In experiments with a fixed batch size of 56, we compared various contrastive methods based on their top1 and top5 matching accuracy for two modalities. The results in Table 14 indicate that our method, scCMIA, demonstrates robust alignment performance in both top1 and top5 across multiple datasets under this batch size. These findings further confirm the effectiveness of scCMIA.

Table 14: Performance comparison of different methods on multi-omics datasets.

| Methods | Modality | 10X Multiome | | ISSAAC-seq | | SHARE-seq | |
|---|---|---|---|---|---|---|---|
| | | Top1 | Top5 | Top1 | Top5 | Top1 | Top5 |
| GLUE | RNA→ATAC | 0.6732 | 0.9662 | 0.7692 | 0.9775 | 0.1510 | 0.4120 |
| | ATAC→RNA | 0.6282 | 0.9575 | 0.7264 | 0.9705 | 0.1686 | 0.4271 |
| Pamona | RNA→ATAC | 0.0184 | 0.0881 | 0.0163 | 0.0836 | NA | NA |
| | ATAC→RNA | 0.0195 | 0.0927 | 0.0174 | 0.0847 | NA | NA |
| UnionCon | RNA→ATAC | 0.0171 | 0.0817 | 0.0148 | 0.0217 | NA | NA |
| | ATAC→RNA | 0.0080 | 0.0590 | 0.0736 | 0.0953 | NA | NA |
| MMD_MA | RNA→ATAC | 0.0430 | 0.1791 | 0.0676 | 0.2676 | NA | NA |
| | ATAC→RNA | 0.0304 | 0.1642 | 0.0633 | 0.2697 | NA | NA |
| scCMIA | RNA→ATAC | **0.7146** | **0.9855** | **0.8595** | **0.9958** | **0.6073** | **0.9552** |
| | ATAC→RNA | **0.7191** | **0.9840** | **0.8632** | **0.9967** | **0.6137** | **0.9548** |

### B.1.4 CROSS-DATASET ZERO-SHOT EXPERIMENTS

In order to apply the pre-trained model in a realistic scenario, we performed the zero-shot task. The experiment was set up as a zero-shot experiment on SNARE-seq (Cortex tissue of mouse) with data trained on SHARE-seq (Skin tissue of mouse). And we used the results obtained by Pamona, UnionCon, and MMD_MA trained on SNARE-seq data as Baseline. The experimental results are shown in Fig. 7. scCMIA demonstrates optimal matching accuracy on doing the zero-shot task, although there is a gap compared to directly on the original dataset, which may be due to the variability between datasets, tissues, and cell types resulting in the limited migration ability of the model, which requires a larger scale and diversity of data to train the model in order to effectively improve the transfer ability of the model.

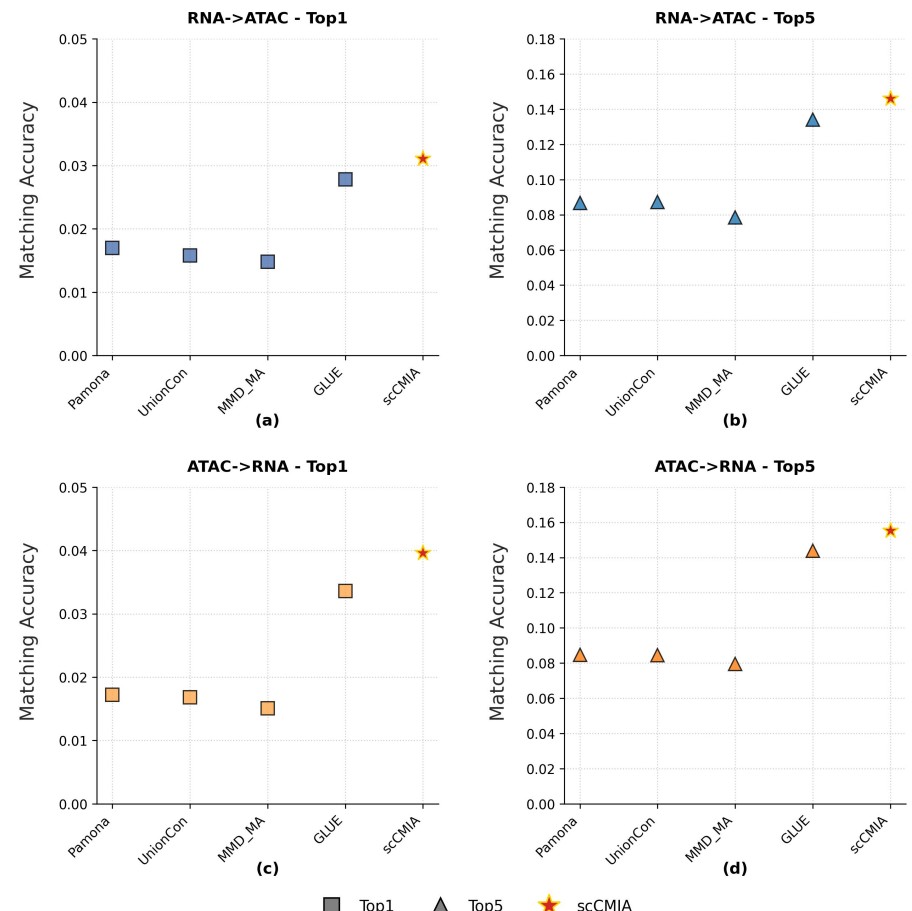

Figure 7: SHARE-seq trained models are made into pair matching zero-shot experiments on the SNARE-seq dataset.

## C USE OF LLMS

We utilized a large language model (LLM) to assist in the writing and editing of this paper. The LLM's role was strictly limited to improving grammar, phrasing, and overall readability. All scientific contributions, including the core ideas, methodology, and interpretation of results, are solely the work of the authors.

