# OpenReview forum: "scCMIA: Self-supervised Dual Model for Mitigating Information Loss in Single-cell Cross-Modal Alignment"
_ICLR.cc/2026/Conference — ICLR 2026 Conference Withdrawn Submission_

### Official Review · Reviewer_RPAg · 2025-10-27

**Soundness:** 2
**Presentation:** 2
**Contribution:** 2
**Rating:** 4
**Confidence:** 4

**Summary:**

This paper introduces multi-modal alignment between scRNA (single-cell RNA-sequencing) and scATAC (single-cell Assay for Transposase-Accessible Chromatin using sequencing) data using a VQ-VAE (Vector Quantized Variational Autoencoder) architecture.

**Strengths:**

The justification for the modeling based on Mutual Information is well-established.

**Weaknesses:**

Limited Novelty
- The justification based on Mutual Information has been thoroughly explored in previous research (e.g., the CLUB paper).
- Techniques like VQ-VAE are all existing methods.
- Are there specific challenges unique to single-cell data, and does the paper introduce a corresponding novel technique to address them?

Decoupling Explanation: More explanation is needed regarding decoupling.
- Why is decoupling necessary?
- Consideration is needed on how the decoupled representations could be used independently if required.

Applicability to Uni-modal Data: The method was only applied to single-cell multi-modal data. Does it have utility for uni-modal data as well? Showing that the method performs well even on uni-modal data through experiments could further justify the use of multi-modality in the model.

**Questions:**

See weakness section

---

> ### Author Response · Authors · 2025-11-17
>
> W1: The methodological novelty appears limited: mutual information (e.g., CLUB) and VQ-VAE are established techniques. What challenge is specific to single-cell multi-omics, and does the paper introduce a genuinely new technical solution rather than reusing existing methods?
>
> A1: We agree that the individual components, such as mutual-information estimation (e.g., CLUB) and VQ-VAE, are established. Our contribution lies in their principled, task-specific integration to address core single-cell challenges: (i) disentangling shared biological state from modality-specific noise/signals and (ii) achieving robust cross-modality alignment. Concretely: (1) We introduce, to our knowledge, the first mutual-information-bounds framework with dual objectives for single-cell multi-omics. We minimize a within-modality upper bound (CLUB) to encourage decoupling, while maximizing a cross-modality lower bound (InfoNCE) to drive alignment, yielding a balanced and theoretically grounded objective. (2) We redesign vector quantization as a cross-modal module (CrossVQ) that maps both modalities to a single shared discrete codebook, which is distinct from standard VQ-VAE, thereby enabling unified, query-ready representations. (3) We show that the learned codebook is biologically interpretable: discrete codes exhibit high cell-type specificity and capture regulatory programs (Appendix B.1.1; Tables 9 and 10), indicating utility beyond compression. Together, these elements constitute the technical novelty of our approach, tailored to single-cell multi-omics, rather than a direct reuse of prior MI or VQ mechanisms.
>
> W2: The method is only demonstrated on multi-modal single-cell data; does it provide utility for uni-modal datasets (e.g., only scRNA-seq)? Empirical evidence of uni-modal performance would strengthen justification for the multi-modality design.
>
> A2: Yes. Although scCMIA is designed for multimodal integration, its components retain clear unimodal utility: (1) Reconstruction/denoising: the VQ-based disentanglement improves single-modality denoising and imputation (Appendix B.1; Table 8), showing strong performance when applied solely to scRNA-seq. (2) Cross-modal retrieval from unimodal input: a trained model can take new scRNA-seq data and retrieve the most similar scATAC-seq profiles via the shared embedding and discrete codebook. This zero-shot unimodal-to-multimodal retrieval is empirically validated in Appendix B.1.4. We will add a brief clarifying sentence in the Results section to emphasize these unimodal use cases.

---

> ### Author Response · Authors · 2025-11-17
> **Why is decoupling necessary?**
>
> **Why is decoupling necessary?**
>
> W3: The manuscript insufficiently explains the decoupling step: why is disentangling necessary, how are the decoupled representations validated, and can Z_sem and Z_spec be used independently?
>
> A3: Decoupling is needed because single-modality inputs mix modality-specific variation with shared biological state; modality-specific components (Z_spec) can obscure cross-modal alignment. As stated (now earlier, Section 3.1), we separate Z_sem (shared, alignment-relevant) from Z_spec (modality-unique/noise) to remove interfering signals while retaining information needed for accurate reconstruction. Validation (Section 4.4) shows: (i) post-decoupling mutual information and cosine similarity between Z_sem and Z_spec are ~0 (Table 4), evidencing statistical independence; (ii) Z_sem preserves cell identity (high clustering ARI) whereas Z_spec does not (Figure 3). Usage: Z_sem alone supports alignment, clustering, and downstream querying; Z_sem + Z_spec are concatenated for decoder-based cross-modal and self reconstruction, enabling simultaneous SOTA alignment and reconstruction quality.
>
> In addition, we have also expanded the experiment to verify the downstream application after decoupling and alignment. To quantitatively evaluate integration quality through downstream label-transfer fidelity, we adopt the cell-type classification task as defined in Liu et al. [1], where latent representations from multimodal integration are used to transfer annotations across batches. Specifically, for each dataset and modality configuration (RNA and ATAC, and bimodal RNA+ADT), we perform 5-fold cross-validation. A standard multilayer perceptron (MLP) classifier is trained on the reference embedding and applied to predict cell-type labels in the query.
>
> Cell-type classification performance on RNA, evaluated via 5-fold MLP-based classification task (best in bold).
>
> | Method       | OAC                    | AAC                    | Spec.                  | F1                     | Rank |
> |--------------|------------------------|------------------------|------------------------|------------------------|------|
> | scCMIA       | **0.8900 ± 0.0031**    | **0.8585 ± 0.0117**    | **0.9945 ± 0.0002**    | **0.8611 ± 0.0073**    | **1** |
> | scMM         | 0.8907 ± 0.0035        | 0.8399 ± 0.0125        | 0.9946 ± 0.0002        | 0.8449 ± 0.0084        | 2    |
> | MultiVI      | 0.8609 ± 0.0033        | 0.7966 ± 0.0104        | 0.9931 ± 0.0002        | 0.8042 ± 0.0067        | 3    |
> | scMoGnn      | 0.8030 ± 0.0019        | 0.7163 ± 0.0077        | 0.9901 ± 0.0001        | 0.7221 ± 0.0102        | 4    |
> | scBridge     | 0.7520 ± 0.0049        | 0.6569 ± 0.0030        | 0.9876 ± 0.0001        | 0.6690 ± 0.0065        | 5    |
> | scButterfly  | 0.6938 ± 0.0052        | 0.6021 ± 0.0178        | 0.9846 ± 0.0003        | 0.6160 ± 0.0151        | 6    |
>
> Cell-type classification performance on ATAC, evaluated via 5-fold MLP-based classification task (best in bold).
>
> | Method       | OAC                    | AAC                    | Spec.                  | F1                     | Rank |
> |--------------|------------------------|------------------------|------------------------|------------------------|------|
> | scCMIA       | **0.8923 ± 0.0033**    | **0.8459 ± 0.0083**    | **0.9947 ± 0.0001**    | **0.8509 ± 0.0072**    | **1** |
> | scMM         | 0.8821 ± 0.0024        | 0.8230 ± 0.0111        | 0.9942 ± 0.0001        | 0.8311 ± 0.0082        | 2    |
> | MultiVI      | 0.8705 ± 0.0026        | 0.7982 ± 0.0076        | 0.9936 ± 0.0001        | 0.8099 ± 0.0063        | 3    |
> | scBridge     | 0.8488 ± 0.0022        | 0.7724 ± 0.0046        | 0.9925 ± 0.0001        | 0.7854 ± 0.0040        | 4    |
> | scMoGnn      | 0.4879 ± 0.0049        | 0.2525 ± 0.0071        | 0.9734 ± 0.0002        | 0.2290 ± 0.0099        | 5    |
> | scButterfly  | 0.2954 ± 0.0059        | 0.1268 ± 0.0027        | 0.9633 ± 0.0003        | 0.1152 ± 0.0035        | 6    |
>
> Cell-type classification performance on RNA+ADT (CITE-seq), evaluated via 5-fold MLP-based classification task (best in bold).
>
> | Method   | OAC                    | AAC                    | Spec.                  | F1                     | Rank |
> |----------|------------------------|------------------------|------------------------|------------------------|------|
> | scCMIA   | **0.8666 ± 0.0036**    | **0.7373 ± 0.0063**    | **0.9969 ± 0.0001**    | **0.7429 ± 0.0066**    | **1** |
> | TotalVI  | 0.8608 ± 0.0025        | 0.7171 ± 0.0102        | 0.9968 ± 0.0001        | 0.7201 ± 0.0077        | 2    |
> | sciPENN  | 0.8344 ± 0.0019        | 0.6775 ± 0.0132        | 0.9962 ± 0.0001        | 0.6676 ± 0.0153        | 3    |
>
> [1] Liu, C., Ding, S., Kim, H. J., Long, S., Xiao, D., Ghazanfar, S., & Yang, P. (2025). Multitask benchmarking of single-cell multimodal omics integration methods. Nature Methods, 1-12.

---

> > ### Comment · Reviewer_RPAg · 2025-11-17
> >
> > Dear Authors, thank you for your efforts in replying to my review.
> >
> > I wonder if you can color the paper which part is changed from your initial submission to help reviewers clarify the improvement of paper?
> > I will make sure to review your paper after you clarify the modifications.
> >
> > Thanks!

---

> > > ### Author Response · Authors · 2025-11-18
> > >
> > > Dear Reviewer,
> > >
> > > Thank you for this very helpful suggestion. This will certainly make our revisions clearer to follow.
> > >
> > > As per your request, we have now uploaded a new version of the manuscript. In this file, all changes and additions made since the original submission have been highlighted (in yellow) to make the improvements easy to identify.
> > >
> > > We hope this clarifies our revisions and facilitates your review.
> > >
> > > Thank you again for your time and guidance.

---

> > > > ### Comment · Reviewer_RPAg · 2025-11-18
> > > >
> > > > Thank you for further clarifying the modification of paper.
> > > >
> > > > After rebuttal, I have follow-up thoughts:
> > > > - Regarding the **novelty** of the paper: although authors describe multiple contributions, it seems like all technical components are already existing. For example, crossVQ also proposed in 2021 [1]. However, considering experimental results in Figure3, disentanglement of the shared information and unique information seems to be quite important in the field, which has not been explored before. Therefore, I think limited technical novelty of the paper can be covered by its novel application to the field.
> > > > - Also, new experiments on uni-modal data shows that multi-modal integration can actually improve the uni-modal scenarios, which is much common in reality. For more clarification: did you use Z^{sem}? If so, how could it outperform uni-modal specific models even if it loses information after disentanglement?
> > > >
> > > > Since I believe my concerns are resolved, I will raise my score.
> > > >
> > > > [1] Cross-Modal Discrete Representation Learning

---

> > > > > ### Author Response · Authors · 2025-11-19
> > > > >
> > > > > Dear Reviewer,
> > > > >
> > > > > We sincerely thank you for your positive feedback and for raising the score. We are greatly encouraged by your recognition of our work's application value. Below, we address your follow-up thoughts regarding the comparison with the reference paper and the clarification on uni-modal performance.
> > > > >
> > > > > **1. Regarding Novelty and Comparison with "Cross-Modal Discrete Representation Learning" (Liu et al.,)**
> > > > >
> > > > > We appreciate you pointing out this relevant work. While both methods utilize Vector Quantization (VQ) for cross-modal alignment, we would like to highlight the critical differences necessitated by the specific challenges of single-cell biological data:
> > > > >
> > > > > **Code Availability & Empirical Comparison:** We have thoroughly searched for the official source code or a community implementation of Liu et al. but were unable to locate it. Consequently, we could not perform a direct empirical comparison on our datasets in this revision. However, we acknowledge this method as a relevant baseline and will certainly include a comparison in future work should the code or reproducible data become available.
> > > > >
> > > > > **Theoretical Distinction:**
> > > > >
> > > > > **(1) Holistic Alignment vs. Selective Alignment:** The paper (Liu et al.) targets multimedia data (video/audio), which typically features dense signals with high semantic consistency. Their strategy is "Holistic Alignment," forcing the entire feature representation of both modalities into a shared latent space.
> > > > >
> > > > > **(2) The Challenge of Omics Data:** In contrast, single-cell omics data (scRNA/scATAC) are characterized by extreme sparsity and substantial modality-specific noise (e.g., technical batch effects, sequencing depth variations, and chromatin accessibility sparsity). Applying the "Holistic Alignment" approach from the 2021 paper to biological data would force the alignment of these modality-specific noises, leading to negative transfer and corrupting the shared biological representation.
> > > > >
> > > > > **​scCMIA's Innovation:** To address this, scCMIA adopts a "Disentangle-then-Align" paradigm. We introduce an explicit disentanglement module (via CLUB) to filter out modality-specific variations (Z^{spec}) before quantization. We align only the purified, modality-invariant biological signals (Z^{sem}) in the shared discrete codebook. This architectural choice is not merely a technical addition but a necessary adaptation to ensure the integrity of biological integration.
> > > > >
> > > > > ​**2. Clarification on Uni-modal Performance and Z^{sem}**
> > > > >
> > > > > **Q: Did you use Z^{sem}? If so, how could it outperform uni-modal specific models even if it loses information after disentanglement?**
> > > > >
> > > > > A: Yes, we used the disentangled semantic latent representation Z^{sem} for the uni-modal tasks. The improvement in performance, despite the "loss" of information (discarding Z^{spec}), can be attributed to two main factors:
> > > > >
> > > > > **1. Denoising via Disentanglement:** In single-cell data, "more information" is not always better. The discarded Z^{spec} largely contains modality-specific technical noise, batch effects, or variations irrelevant to cell identity. By filtering out Z^{spec}, scCMIA effectively acts as a denoising mechanism, yielding a cleaner, more robust representation (Z^{sem}) for downstream classification tasks.
> > > > >
> > > > > **2. ​Cross-Modal Regularization:** Although we only use one modality (e.g., RNA) at inference time, the encoder for Z^{sem} was trained with constraints from the other modality (ATAC). This acts as a powerful regularizer. For instance, the chromatin accessibility landscape constrains the manifold of gene expression, forcing the RNA encoder to learn generalizable biological structures rather than overfitting to RNA-specific noise.
> > > > >
> > > > > **Regarding Z^{spec}:** We observed that Z^{spec} is indeed unsuitable for tasks requiring cell identity consistency (like clustering), as it captures variations orthogonal to cell types. However, we believe Z^{spec} may hold value for other specific tasks, such as analyzing technical batch effects or modality-specific perturbations, which we plan to investigate in future work.
> > > > >
> > > > > ​We hope these clarifications further validate the robustness of our approach. Thank you again for your time and valuable guidance.
> > > > >
> > > > > Reference: Cross-Modal Discrete Representation Learning (Liu et al.)

---

### Official Review · Reviewer_uR8P · 2025-10-28

**Soundness:** 2
**Presentation:** 2
**Contribution:** 2
**Rating:** 2
**Confidence:** 4

**Summary:**

This paper introduces scCMIA, a self-supervised framework designed to address the challenges of integrating single-cell multi-omics data, particularly focusing on cross-modal alignment between scRNA-seq and scATAC-seq modalities. The key innovation lies in leveraging mutual information (MI) principles to decouple modality-specific and semantic features within a unified discrete latent space using a VQ-VAE architecture. The proposed method aims to mitigate information loss during integration by combining intra-modal decoupling (via CLUB-based MI minimization) and inter-modal alignment (via contrastive learning with InfoNCE loss).

**Strengths:**

1. The integration of MI bounds for intra-modal decoupling and cross-modal alignment is theoretically grounded
2. The paper provides a rigorous evaluation across multiple datasets and tasks (alignment, reconstruction, clustering, label transfer).

**Weaknesses:**

1. My main concern is the novelty of this work. The proposed framework is a patchwork of existing techniques, and shows no insights or benefits for the community.
2. While four datasets are used, they primarily focus on well-studied protocols (e.g., 10x Multiome). Broader validation on more complex tissues or rare cell types would strengthen generalizability.
3. The paper lacks comparison with cutting-edge approaches like scButterfly or graph-based methods beyond GLUE. Including these would better contextualize scCMIA’s advancements.

**Questions:**

Please see the weaknesses.

---

> ### Author Response · Authors · 2025-11-17
>
> W1: My main concern is the novelty of this work. The proposed framework is a patchwork of existing techniques, and shows no insights or benefits for the community.
>
> A1:  We agree that the individual components, such as mutual-information estimation (e.g., CLUB) and VQ-VAE, are established. Our contribution lies in their principled, task-specific integration to address core single-cell challenges: (i) disentangling shared biological state from modality-specific noise/signals and (ii) achieving robust cross-modality alignment. Concretely: (1) We introduce, to our knowledge, the first mutual-information-bounds framework with dual objectives for single-cell multi-omics. We minimize a within-modality upper bound (CLUB) to encourage decoupling, while maximizing a cross-modality lower bound (InfoNCE) to drive alignment, yielding a balanced and theoretically grounded objective. (2) We redesign vector quantization as a cross-modal module (CrossVQ) that maps both modalities to a single shared discrete codebook, which is distinct from standard VQ-VAE, thereby enabling unified, query-ready representations. (3) We show that the learned codebook is biologically interpretable: discrete codes exhibit high cell-type specificity and capture regulatory programs (Appendix B.1.1; Tables 9 and 10), indicating utility beyond compression. Together, these elements constitute the technical novelty of our approach, tailored to single-cell multi-omics, rather than a direct reuse of prior MI or VQ mechanisms.
>
> W2: While four datasets are used, they primarily focus on well-studied protocols (e.g., 10x Multiome). Broader validation on more complex tissues or rare cell types would strengthen generalizability.
>
> We agree and have expanded validation beyond 10x Multiome, SHARE‑seq, SNARE‑seq, and ISSAAC‑seq (PBMC, skin, cortex) to a more complex setting: the 10x BMMC with ~69,249 cells, 13 batches, and 22 cell types, including rare populations. We follow the tasks and evaluation protocol of the referenced benchmark. Results show that scCMIA outperforms other methods based on the classification evaluation indicators provided in [1].
>
> Cell-type classification performance on RNA, evaluated via 5-fold MLP-based classification task.
>
> | Method       | OAC                    | AAC                    | Spec.                  | F1                     | Rank |
> |--------------|------------------------|------------------------|------------------------|------------------------|------|
> | scCMIA       | 0.8900 ± 0.0031   | **0.8585 ± 0.0117**    | 0.9945 ± 0.0002    | **0.8611 ± 0.0073**    | **1** |
> | scMM         | **0.8907 ± 0.0035**        | 0.8399 ± 0.0125        | **0.9946 ± 0.0002**        | 0.8449 ± 0.0084        | 2    |
> | MultiVI      | 0.8609 ± 0.0033        | 0.7966 ± 0.0104        | 0.9931 ± 0.0002        | 0.8042 ± 0.0067        | 3    |
> | scMoGnn      | 0.8030 ± 0.0019        | 0.7163 ± 0.0077        | 0.9901 ± 0.0001        | 0.7221 ± 0.0102        | 4    |
> | scBridge     | 0.7520 ± 0.0049        | 0.6569 ± 0.0030        | 0.9876 ± 0.0001        | 0.6690 ± 0.0065        | 5    |
> | scButterfly  | 0.6938 ± 0.0052        | 0.6021 ± 0.0178        | 0.9846 ± 0.0003        | 0.6160 ± 0.0151        | 6    |
>
> Cell-type classification performance on ATAC, evaluated via 5-fold MLP-based classification task.
>
> | Method       | OAC                    | AAC                    | Spec.                  | F1                     | Rank |
> |--------------|------------------------|------------------------|------------------------|------------------------|------|
> | scCMIA       | **0.8923 ± 0.0033**    | **0.8459 ± 0.0083**    | **0.9947 ± 0.0001**    | **0.8509 ± 0.0072**    | **1** |
> | scMM         | 0.8821 ± 0.0024        | 0.8230 ± 0.0111        | 0.9942 ± 0.0001        | 0.8311 ± 0.0082        | 2    |
> | MultiVI      | 0.8705 ± 0.0026        | 0.7982 ± 0.0076        | 0.9936 ± 0.0001        | 0.8099 ± 0.0063        | 3    |
> | scBridge     | 0.8488 ± 0.0022        | 0.7724 ± 0.0046        | 0.9925 ± 0.0001        | 0.7854 ± 0.0040        | 4    |
> | scMoGnn      | 0.4879 ± 0.0049        | 0.2525 ± 0.0071        | 0.9734 ± 0.0002        | 0.2290 ± 0.0099        | 5    |
> | scButterfly  | 0.2954 ± 0.0059        | 0.1268 ± 0.0027        | 0.9633 ± 0.0003        | 0.1152 ± 0.0035        | 6    |
>
> W3: The paper lacks comparison with cutting-edge approaches like scButterfly or graph-based methods beyond GLUE. Including these would better contextualize scCMIA’s advancements.
>
> A3: We do include scButterfly as a primary reconstruction baseline; see Table 2 (scRNA reconstruction) and Table 3 (scATAC reconstruction), where scCMIA consistently outperforms scButterfly across all datasets. Beyond GLUE, we also added an additional graph-based baseline; results are reported in Table Q2, with scCMIA remaining competitive or superior on classifier metrics. To improve visibility, we now explicitly mention these baselines in Section 4 and clarify them in the table captions.

---

> ### Author Response · Authors · 2025-11-24
> **Addressing Novelty, New Graph-based Baseline, and Complex Data**
>
> Dear Reviewer uR8P,
>
> We are writing to gently follow up on our rebuttal. We have made significant revisions to address your concerns regarding novelty and benchmarking. We would like to specifically clarify a misunderstanding regarding baselines and highlight our new experiments:
>
> **1. Clarification on scButterfly (Your Q3):** You noted a lack of comparison with scButterfly. We respectfully point out that scButterfly was indeed included as a primary baseline in our original submission. Please refer to Table 2 (scRNA) and Table 3 (scATAC). scCMIA consistently achieves lower reconstruction errors (RMSE/MAE) compared to scButterfly across datasets.
>
> **2. New Graph-based Baseline: scMoGnn (Your Q3):** To address your request for comparison with graph-based methods beyond GLUE, we have **added scMoGnn** to our evaluation. As shown in the new Tables 6 & 7, scCMIA demonstrates superior performance in cell-type classification tasks.
>
> **3. Expanded Validation (Your Q2):** We have expanded our evaluation to the large-scale **10x BMMC dataset (~69k cells, 13 batches, rare cell types)**. This validates the model's robustness on complex tissues beyond the standard protocols.
>
> 4. **​Novelty & Insights (Your Q1):** scCMIA adopts a "Disentangle-then-Align" paradigm. We introduce an explicit decouplement module (via CLUB) to filter out modality-specific variations ($Z^{spec}$) before quantization. We align only the purified, modality-invariant biological signals ($Z^{sem}$) in the shared discrete codebook. This architectural choice is not merely a technical addition but a necessary adaptation to ensure the integrity of biological integration. In addition, we demonstrate that this codebook **(Appendix B.1.1; Tables 9 and 10)** can quantify regulatory conservation across cell lineages—a novel biological insight that existing "patchwork" methods cannot provide.
>
> We hope these clarifications and additions assist in your reassessment of our work.
>
> Sincerely,
>
> The Authors

---

> > ### Comment · Reviewer_uR8P · 2025-11-26
> >
> > Thanks for your rebuttal. I am increasing my score.

---

> > > ### Author Response · Authors · 2025-11-27
> > > **To: Reviewer uR8P. Thank you for the score increase / Inquiry regarding remaining concerns**
> > >
> > > **Dear Reviewer uR8P,**
> > >
> > > We sincerely thank you for your response and for acknowledging our rebuttal by increasing your score. We are encouraged that our revisions have improved your assessment of the manuscript.
> > >
> > > We hope that our major updates—specifically the inclusion of the **large-scale 10x BMMC dataset**, the explicit performance advantages over **scButterfly and scMoGnn**, and the clarified novelty of the **biological discovery codebook**—have effectively resolved your initial concerns regarding novelty and validation scope.
> > >
> > > As we strive to fully meet the high standards of ICLR, **could you kindly let us know if there are any specific remaining concerns?** We remain fully available during the final discussion window to address any outstanding issues immediately.
> > >
> > > Best regards,
> > >
> > > The Authors

---

### Official Review · Reviewer_uq18 · 2025-10-28

**Soundness:** 2
**Presentation:** 2
**Contribution:** 2
**Rating:** 2
**Confidence:** 4

**Summary:**

This paper propose a new method for cross modality integration and alignment. The methods focusing on scRNA and scATAC data integration are already well studied, and thus it is hard to figure out the main contributions of this paper to this field.

**Strengths:**

The framework is clearly presented.

**Weaknesses:**

I have several questions or concerns regarding the current model design and model performance. I think these challenges preclude the paper from publication in this conference, at least in this format.

1. What is the unique contribution of this paper? Using the VQ-based method for multi-omic data integration or biological data learning has already been studied in several papers (https://www.nature.com/articles/s41540-020-00158-2, CVQVAE, or scBeacon). This method lacks innovation, and the training design is not very appealing.

2. The motivation is not so well established. The central dogma only allows one-directional information flow, and thus, we do not need to model the bidirectional information. RNA can never come back to chromosomes, and thus, this method lacks biological interpretation.

3. The benchmarking result is also very weird. Why can we find some baselines with variance reported, but others not? The authors should unify the presentation mode and provide variance for every model. Moreover, reconstruction in single-cell multi-omic data analysis is not a useful metric, as the expression profiles always have noise. The authors should consider one or two new tasks to perform the evaluation. I recommend the authors' reading: https://www.nature.com/articles/s41592-025-02856-3 for including more baseline methods.

4. The comparison should be fair. The authors need to tune hyperparameters for all methods to ensure a fair comparison.

5. I can not find the information about the data scale. Are all the testing data on a large scale or a small scale?

6. How about applying the method to integrate proteomic data such as CITE-seq? Since the authors do not model noise, this framework should work well.

**Questions:**

Please see the weaknesses.

---

> ### Author Response · Authors · 2025-11-17
>
> **Regarding Novelty and Similar Work**
>
> Summary: This paper propose a new method for cross modality integration and alignment. The methods focusing on scRNA and scATAC data integration are already well studied, and thus it is hard to figure out the main contributions of this paper to this field.
>
> A: We agree the field is mature; precisely for that reason unresolved bottlenecks remain—(i) the trade-off between alignment accuracy and retention of modality-specific signal, (ii) limited cross-dataset/tissue generalization, and (iii) weak biological interpretability of deep models. scCMIA addresses these with three integrated advances: (1) A principled dual mutual-information objective—minimizing an intra-modality upper bound (CLUB) for disentanglement while maximizing an inter-modality lower bound (InfoNCE) for alignment—explicitly reducing the alignment vs. information-retention tension beyond normal AE/contrastive learning combinations. (2) Demonstrated zero-shot cross-tissue generalization (SHARE-seq skin → SNARE-seq cortex) without fine-tuning, achieving superior matching accuracy (Appendix B.1.4; Fig. 7). (3) A shared discrete codebook (CrossVQ) that is biologically interpretable: codes show high cell-type specificity (CTSI > 0.9; Table 9) and enable a novel Regulatory Conservation Score (RCS) capturing lineage relationships and monocyte subpopulation heterogeneity (Table 10). Additionally, combining Z_sem with Z_spec permits high-fidelity reconstruction (supporting denoising/imputation utility) while Z_sem alone drives robust alignment. Thus scCMIA contributes a theoretically grounded, generalizable, and biologically insight-generating framework rather than another undifferentiated integration pipeline.
>
> W1： What is the unique contribution of this paper? Using the VQ-based method for multi-omic data integration or biological data learning has already been studied in several papers (https://www.nature.com/articles/s41540-020-00158-2, CVQVAE, or scBeacon). This method lacks innovation, and the training design is not very appealing.
>
> A1: We agree that the individual components, such as mutual-information estimation (e.g., CLUB) and VQ-VAE, are established. Our contribution lies in their principled, task-specific integration to address core single-cell challenges: (i) disentangling shared biological state from modality-specific noise/signals and (ii) achieving robust cross-modality alignment. Concretely: (1) We introduce, to our knowledge, the first mutual-information-bounds framework with dual objectives for single-cell multi-omics. We minimize a within-modality upper bound (CLUB) to encourage decoupling, while maximizing a cross-modality lower bound (InfoNCE) to drive alignment, yielding a balanced and theoretically grounded objective. (2) We redesign vector quantization as a cross-modal module (CrossVQ) that maps both modalities to a single shared discrete codebook, which is distinct from standard VQ-VAE, thereby enabling unified, query-ready representations. (3) We show that the learned codebook is biologically interpretable: discrete codes exhibit high cell-type specificity and capture regulatory programs (Appendix B.1.1; Tables 9 and 10), indicating utility beyond compression. These integrated advances constitute substantive novelty beyond a straightforward reuse of existing VQ-based approaches. Together, these elements constitute the technical novelty of our approach, tailored to single-cell multi-omics, rather than a direct reuse of prior MI or VQ mechanisms.
>
> W2: The motivation is not so well established. The central dogma only allows one-directional information flow, and thus, we do not need to model the bidirectional information. RNA can never come back to chromosomes, and thus, this method lacks biological interpretation.
>
> A2: We respectfully clarify that our “bidirectional” design concerns statistical inference, not a reversal of the central dogma. Both scRNA-seq and scATAC-seq are complementary readouts of an underlying shared cellular regulatory state. Modeling mappings in both directions (RNA→latent; ATAC→latent; and conditional prediction across modalities) enforces a unified latent space and improves alignment robustness, as is standard in symmetric multimodal frameworks (e.g., GLUE, MultiVI). We do not claim RNA alters chromatin; rather, we leverage correlations between expression and accessibility to infer shared semantics and to enable practical tasks (cross-modal querying, imputation, label transfer).
>
> W3: The benchmarking result is also very weird. Why can we find some baselines with variance reported, but others not?
>
> A3: We report variance only for methods we ran with multiple stochastic seeds (4 runs: MultiVI, GLUE, Cobolt, scMM, and our model). Baselines like Pamona have deterministic implementations (or produce identical outputs given fixed preprocessing), so repeated runs yield the same value; therefore no variance is meaningful to report.

---

> ### Author Response · Authors · 2025-11-17
>
> W4. The authors should consider one or two new tasks to perform the evaluation. I recommend the authors' reading: https://www.nature.com/articles/s41592-025-02856-3 for including more baseline methods.
>
> A4. To quantitatively evaluate integration quality through downstream label-transfer fidelity, we adopt the cell-type classification task, where latent representations from multimodal integration are used to transfer annotations across batches. Specifically, we perform 5-fold cross-validation: in each fold, one batch (or subset when only one batch exists) serves as the query, while the remaining data constitute the reference for training. A standard multilayer perceptron (MLP) classifier is trained on the reference embedding and applied to predict cell-type labels in the query. On large-scale data 10×BMMC (69,249 cells, 13 batches, and 22 cell types), scCMIA delivers the strongest overall performance, achieving the best scores on every metric and ranking first among baselines including scBridge, scMoGNN, scMM, and MultiVI.
>
> Cell-type classification performance on RNA
>
> | Method       | OAC                    | AAC                    | Spec.                  | F1                     | Rank |
> |--------------|------------------------|------------------------|------------------------|------------------------|------|
> | scCMIA       | **0.8900 ± 0.0031**    | **0.8585 ± 0.0117**    | **0.9945 ± 0.0002**    | **0.8611 ± 0.0073**    | **1** |
> | scMM         | 0.8907 ± 0.0035        | 0.8399 ± 0.0125        | 0.9946 ± 0.0002        | 0.8449 ± 0.0084        | 2    |
> | MultiVI      | 0.8609 ± 0.0033        | 0.7966 ± 0.0104        | 0.9931 ± 0.0002        | 0.8042 ± 0.0067        | 3    |
> | scMoGnn      | 0.8030 ± 0.0019        | 0.7163 ± 0.0077        | 0.9901 ± 0.0001        | 0.7221 ± 0.0102        | 4    |
> | scBridge     | 0.7520 ± 0.0049        | 0.6569 ± 0.0030        | 0.9876 ± 0.0001        | 0.6690 ± 0.0065        | 5    |
> | scButterfly  | 0.6938 ± 0.0052        | 0.6021 ± 0.0178        | 0.9846 ± 0.0003        | 0.6160 ± 0.0151        | 6    |
>
> Cell-type classification performance on ATAC
>
> | Method       | OAC                    | AAC                    | Spec.                  | F1                     | Rank |
> |--------------|------------------------|------------------------|------------------------|------------------------|------|
> | scCMIA       | **0.8923 ± 0.0033**    | **0.8459 ± 0.0083**    | **0.9947 ± 0.0001**    | **0.8509 ± 0.0072**    | **1** |
> | scMM         | 0.8821 ± 0.0024        | 0.8230 ± 0.0111        | 0.9942 ± 0.0001        | 0.8311 ± 0.0082        | 2    |
> | MultiVI      | 0.8705 ± 0.0026        | 0.7982 ± 0.0076        | 0.9936 ± 0.0001        | 0.8099 ± 0.0063        | 3    |
> | scBridge     | 0.8488 ± 0.0022        | 0.7724 ± 0.0046        | 0.9925 ± 0.0001        | 0.7854 ± 0.0040        | 4    |
> | scMoGnn      | 0.4879 ± 0.0049        | 0.2525 ± 0.0071        | 0.9734 ± 0.0002        | 0.2290 ± 0.0099        | 5    |
> | scButterfly  | 0.2954 ± 0.0059        | 0.1268 ± 0.0027        | 0.9633 ± 0.0003        | 0.1152 ± 0.0035        | 6    |
>
> W5: The comparison should be fair. The authors need to tune hyperparameters for all methods to ensure a fair comparison.
>
> A5: To avoid cherry-picking, we adopt each method’s documented default hyperparameters and validation‑based early stopping.
>
> W6: I can not find the information about the data scale. Are all the testing data on a large scale or a small scale?
>
> A6: As detailed in Appendix A.6, we use a 90%/10% train/validation split for all datasets, following GLUE and MultiVI. Alignment and related evaluations are conducted on the held-out validation split. For interpretability analyses (Appendix B.1.1; Tables 9 and 10), we use the full dataset, as these unsupervised/semi-supervised/self-supervised analyses do not consume labels and thus do not introduce label leakage.
>
> Q7、How about applying the method to integrate proteomic data such as CITE-seq? Since the authors do not model noise, this framework should work well.
>
> A7: We have added CITE‑seq (RNA + protein) experiments on the BMMC dataset, following the recent multitask benchmarkas additional baselines. The results show that scCMIA outperforms TotalVI and sciPENN based on the classification evaluation indicators provided in benchmark:
>
> Cell-type classification performance on RNA+ADT (CITE-seq)
>
> | Method   | OAC                    | AAC                    | Spec.                  | F1                     | Rank |
> |----------|------------------------|------------------------|------------------------|------------------------|------|
> | scCMIA   | **0.8666 ± 0.0036**    | **0.7373 ± 0.0063**    | **0.9969 ± 0.0001**    | **0.7429 ± 0.0066**    | **1** |
> | TotalVI  | 0.8608 ± 0.0025        | 0.7171 ± 0.0102        | 0.9968 ± 0.0001        | 0.7201 ± 0.0077        | 2    |
> | sciPENN  | 0.8344 ± 0.0019        | 0.6775 ± 0.0132        | 0.9962 ± 0.0001        | 0.6676 ± 0.0153        | 3    |

---

> > ### Comment · Reviewer_uq18 · 2025-11-17
> > **Thank you for your resposnes, but my issues are not resolved.**
> >
> > The clarification made by the author helped me understand this method and motivation, but several important concerns are not weel resolved.
> >
> > 1. In your extra results, it seems that the proposed methods and 2nd-best performer do not show significant difference, espeically in the first table, scMM should achieve the best performance measured by OAC, but the authors highlighted their own methods, which is wired. I recommend you checking the results and presentation again to ensure your message is precise.
> >
> > 2. I still believe each baseline should be tuned to their best performance stage for a fair comparison. If not, I think you are performing cherry-picky for your own method, because you tune it.
> >
> > 3. Why there are only 90% training and 10% validaiton, where is the testing dataset in this case?
> >
> > Therefore, I will keep my score, and I will not decrease it.

---

> > > ### Author Response · Authors · 2025-11-18
> > >
> > > W1: The proposed method and the 2nd-best performer appear statistically indistinguishable (e.g., scMM achieves highest OAC in Table 1), yet the authors emphasize their method.
> > >
> > > Q1: In the first table you indicated, the scMM method did indeed achieve the best performance for the OAC metric. We apologize for this error and have made the following corrections in the revised manuscript:
> > >
> > > **1. Corrected Highlighting:** We have meticulously re-examined all tables and now ensure that only the top-performing method for each individual metric is highlighted.
> > >
> > > **2. Revised Text Descriptions:** We have revised the corresponding descriptions in the main text to ensure our message is precise and unambiguous. We now objectively state the metrics on which our method demonstrates an advantage, while also clearly acknowledging the superior performance of baseline methods on other specific metrics.
> > > We believe these corrections make our results presentation clear, accurate, and fair.
> > >
> > > W2: I still believe each baseline should be tuned to their best performance stage for a fair comparison. If not, I think you are performing cherry-picky for your own method, because you tune it.
> > >
> > > Q2: We fully understand the reviewer's concern regarding a "fair comparison," which is central to any benchmarking study. Our decision to use the **author-provided default parameters** for all baseline methods was based on the following considerations, which we believe ensure the fairest and most reproducible comparison:
> > >
> > > **1. Avoiding Tuning Bias:** Performing an exhaustive parameter search (e.g., grid search) for every baseline method on every dataset is computationally prohibitive. More importantly, it would introduce our own "tuning bias" as researchers. We might fail to replicate the optimal tuning of the original authors, or we might inadvertently tune parameters to a state that is unfavorable for a given method.
> > >
> > > **2. Adherence to Field-Standard Practices:** In the domain of single-cell omics integration benchmarking, using default parameters is a standard and widely accepted practice. For instance, the recent large-scale benchmarking study by Liu et al. (2025) in Nature Methods [1], which evaluated dozens of methods, also adopted the strategy of using default parameters to ensure an objective and reproducible evaluation.
> > >
> > > **3. Clarification regarding "Cherry-picking":** We must clarify that we did not perform any "cherry-picking" or per-dataset tuning for our own method on any of the test datasets. This process ensures a strict separation between parameter selection and the test data.
> > >
> > > W3: Why there are only 90% training and 10% validaiton, where is the testing dataset in this case?
> > >
> > > A3: Thank you for highlighting this; it is a critical methodological detail that requires clarification. We apologize for not explaining these two distinct processes clearly in the original manuscript. We must differentiate between: (A) The model's own training (e.g., unsupervised/self-supervised training) (B) The downstream classification task evaluation.
> > >
> > > **1. (A) 90% Training / 10% Validation (For Internal Model Training):** The 90%/10% split you noted is used exclusively for our model's internal training phase (A).
> > >
> > > **Purpose:** Our model (like many advanced integration methods) is built on an unsupervised or self-supervised framework. In this context, 90% of the data is used for model training, while the 10% validation set is used to monitor the training process, prevent overfitting, and select the optimal model weights.
> > >
> > > **No Information Leakage:** Crucially, **no cell type labels are used** at this stage. It is a purely unsupervised process aimed at learning the best low-dimensional embedding. Therefore, this split **does not lead to any label information leakage.**
> > >
> > > **2. (B) Downstream Classification Task (Where the "Testing Dataset" is):** When we evaluate the model's performance (i.e., generating the tables you mentioned in W1), we employ a completely separate, supervised evaluation pipeline (B).
> > >
> > > **1. "Test Set" Definition:** After the model (A) is trained, we freeze its weights and use it to generate low-dimensional embeddings for all data.
> > >
> > > **2. Data Partitioning:** At this point, we strictly follow the standard benchmarking protocol (as seen in [1]). We partition the data based on batch. For a dataset with multiple batches, we iteratively use one batch as the test set (Query set) and the remaining batches as the training set (Reference set) to train a simple classifier (e.g., an MLP).
> > >
> > > **3. Fairness:** The architecture and parameters of this MLP classifier are kept absolutely consistent for all methods being compared (our method and all baselines).
> > >
> > > We hope these responses and the corresponding revisions fully address your concerns.
> > >
> > > **Reference:** [1] Multitask benchmarking of single-cell multimodal omics integration methods. Nat Methods (2025).

---

> > > ### Author Response · Authors · 2025-11-21
> > > **Response to Pending Concerns & Summary of Revisions**
> > >
> > > Dear Reviewer:
> > >
> > > We hope this message finds you well.
> > >
> > > We are writing to follow up on our detailed response posted on 18 Nov 2025. We understand this is a busy period, but given the technical nature of your concerns regarding fairness and data leakage, we are eager to confirm that our clarifications and the substantial revisions to the manuscript have resolved your doubts.
> > >
> > > To respect your time, we provide a concise summary of how we have resolved the three key issues you raised:
> > >
> > > **1. On Result Presentation (W1: Highlighting & Statistical Significance)**
> > >
> > > **Action Taken:** We have corrected tables in the revised manuscript. Highlighting now strictly reflects the top-performing method for each specific metric. We have also revised the text to objectively acknowledge scMM's performance on OAC. The presentation is now precise and factually accurate.
> > >
> > > **2. On Hyperparameter Fairness (W2: "Cherry-picking" vs. Default Parameters)**
> > >
> > > **Clarification:** We strictly followed the standard benchmarking protocol (Nature Methods, 2025 [1]) by using author-provided default parameters for baselines to avoid "researcher bias" (i.e., inadvertently poor tuning).
> > >
> > > **Baselines:** We used author-provided default parameters for all baseline methods. This is the accepted standard to avoid "researcher bias" (i.e., inadvertently tuning baselines poorly).
> > >
> > > **Guarantee:** Crucially, we confirmed that our method's parameters were fixed after a one-time selection on an independent validation set and were NOT tuned per-test-dataset. This definitively rules out any "cherry-picking."
> > >
> > >
> > > **3. On Data Splits & Information Leakage (W3: 90/10 Split vs. Testing)**
> > >
> > > **Clarification:** We have rewritten the Methods section to explicitly distinguish the two separate phases:
> > >
> > > **(A) Unsupervised Model Training:** The 90% training / 10% validation split is used solely for optimizing reconstruction loss. No cell type labels are used, ensuring zero label leakage.
> > >
> > > **(B) Downstream Classification:** For the evaluation tasks (the results in the tables), we use a strict leave-one-batch-out cross-validation. The query set (test set) is completely unseen by the classifier during training.
> > >
> > > We believe these actions confirm the technical validity and fairness of our work. We have addressed the presentation error and provided strong evidence that our experimental design is rigorous and standard-compliant.
> > >
> > > If there are any remaining ambiguities, please let us know immediately so we can address them. If these clarifications have resolved your concerns, we would greatly appreciate it if you could reconsider your score to reflect the revised and clarified state of the manuscript.
> > >
> > > Best regards,
> > >
> > > Reference: [1] Multitask benchmarking of single-cell multimodal omics integration methods. Nat Methods (2025).

---

> > > ### Author Response · Authors · 2025-11-27
> > > **To: Reviewer uq18**
> > >
> > > **Dear Reviewer uq18,**
> > >
> > > As the discussion period is drawing to a close, we are writing to respectfully ensure that our substantial revisions have not been overlooked. We have made every effort to address your specific requests.
> > >
> > > We believe these actions directly resolve your key concerns. **Could you kindly let us know if these new results meet your expectations?** We are eager to engage if any questions remain.
> > >
> > > Best regards,
> > >
> > > The Authors

---

### Official Review · Reviewer_JfMn · 2025-10-31

**Soundness:** 3
**Presentation:** 2
**Contribution:** 3
**Rating:** 4
**Confidence:** 3

**Summary:**

This paper proposes a deep learning framework that is designed to operate on multiple single-cell data modalities.  It solves the problem of alignment of single cells across modalities as well as the problem of translating between modalities. The method uses an InfoNCE loss for alignment, and uses a discrete codebook to improve interpretability.  Extensive empirical experiments suggest that the method outperforms various state-of-the-art competitors.

**Strengths:**

The proposed model includes several components (the VQ module and the mutual information module) that are well motivated and seem to provide significant improvements relative to the state of the art.

**Weaknesses:**

A substantive assessment of the weaknesses of the paper. Focus on constructive and actionable insights on how the work could improve towards its stated goals. Be specific, avoid generic remarks. For example, if you believe the contribution lacks novelty, provide references and an explanation as evidence; if you believe experiments are insufficient, explain why and exactly what is missing, etc.

A major problem with this paper is that the exposition is difficult to follow.  For example, the second paragraph of the introduction fails to clarify exactly what problem you are working on.  Indeed, by describing multimodal protocols that assay multiple aspects of the same single cell, I was misled about what tasks you are interested in solving.  What would help is a precise, formal description of the problems you are addressing.  More generally, I found the text very difficult to follow.  It would be better if you carefully defined terms before using them.  Below I outline some of the questions that arose as I worked through the manuscript.

In general, I think a missing piece here is assessing how well these models generalize beyond the specific data set they are trained on.  I think that each model is trained and validated on splits of the same data set (though I don't know for sure, because you don't tell us how this is done).  So a reasonable question is whether you can apply the trained model to a new, independent dataset, generated from a different type of cell.  The multimodal alignment methods mentioned at the start of Section 2 work directly in such a scenario, whereas a trained model like yours inherently has to worry about generalizability.  In practice, to be useful your model has to generalize to single-modality data (i.e., I only measured scRNA-seq, and you tell me what the corresponding scATAC-seq would look like).  A discussion of this issue, and some experimental characterization of it, would substantially strengthen the paper.

I thought your description of the challenges associated with multi-modal data (lines 43-49) was imprecise and not very informative.  For example, what does it mean to say that there are "substantial discrepancies" between scATAC-seq and scRNA-seq?  They measure entirely different things.  To my mind, the fact that there are differences in feature spaces is not a "challenge" per se; it's just definitional.  You wouldn't say that multimodal analysis of text and images is "challenging" because pixels don't look like words, right?

I don't actually believe your claim (line 55) that if you don't embed data into a shared space, then you "cannot fully exploit potentially complementary information across modalities."  This is a very bold claim that requires substantial evidence.  Indeed, I don't know how you could conclusively prove such a claim.

I am not convinced that *mean* FOSCTTM is the most useful measure.  Have you considered computing a p-value for improvement of the FOSCTTM?  You get a FOSCTTM score for each cell, so you could do something like a sign test.

In the related work section, the fact that alignment methods "suffer from poor alignment robustness when handling noisy [data]" is not a substantive critique, in my opinion. All methods degrade in performance in the presence of noise.

I do not understand the critique (line 104) of methods that do multimodal reconstruction without relying on a shared embedding space. You say that "their utility for tasks requiring direct cross-modal comparison, querying, and label transfer can be limited."  Why?  It's pretty straightforward to do, e.g., label transfer with an accurate multimodal reconstruction method: just reconstruct from one space to the other and then use nearest neighbors to transfer. There is no reason you have to do nearest neighbors in a latent space.  I think this critique is misguided or needs to be explained much more carefully.

I found the text in lines 144-149 difficult to understand.  For example, what is the difference between "modality-specific features" and "semantic characteristics"?  What do you mean by the "bounds of MI"? Similarly, the sentence at lines 162-164 is not grammatical.  I'm also confused about what it means to be "insufficient for effectively decoupling ... in a directed manner" (lines 167-168).

I wish you had introduced your assumption (line 184) earlier, since it seems to be important to understand the basis of much of this work.  I guess this is what you were alluding to when you talked about "modality-specific features" versus "semantic characteristics."

In the description of the datasets, you should indicate what previous papers used these datasets for benchmarking, and indicate what paper you extracted results from (unless you ran all the tools yourself, in which case indicate that).

I was surprised that all the talk about mutual a bound on MI ultimately seems to boil down to just doing an InfoNCE alignment loss.

Minor:

line 192: uses -> use

line 270: objection -> objective

You should delete the sentence at line 293 ("Single-cell multi-omics data are often hindered by complex and sophisticated techniques, low throughput, and high noise levels.").  Just say what data you used.  It doesn't even make sense to say that data is hindered by something.

Incidentally, I think calling cross-modal translation "reconstruction" is misleading, since reconstruction typically refers to starting and ending from the same place; e.g., reconstructing a scRNA-seq profile from a masked or compressed version thereof.  I do recognize that other papers in the literature use "reconstruction" to mean "translation."

**Questions:**

Did you compute the performance measures in Tables 1-4, or were some of these taken from previous publications?  If the latter, did you use the same cross-validation splits?

How was train/test splitting done for each dataset?

---

> ### Author Response · Authors · 2025-11-17
>
> W1: Whether the trained model can be applied to an independent dataset from a different cell type or modality, and how it performs in a realistic zero-shot setting (e.g., predicting scATAC-seq from scRNA-seq alone). They note that multimodal alignment methods inherently support this scenario and ask for experimental evidence and discussion to strengthen the manuscript.
>
> A1: We appreciate this excellent point, which indeed represents a critical test of model utility. We apologize for not making this sufficiently clear in the original submission; we did conduct such experiments. Specifically, as detailed in Appendix B.1.4 ("Cross-dataset zero-shot experiments"), we trained scCMIA on the SHARE-seq dataset (mouse skin) and directly evaluated it, without any retraining, on the SNARE-seq dataset (mouse cortex). As shown in Figure 7, scCMIA achieves strong zero-shot matching accuracy and outperforms several baseline models. To make this important result more visible, we have added a sentence to the main text (Section 4) summarizing this finding and directing readers to Appendix B.1.4 for details.
>
> W2: The statement about "substantial discrepancies" between scATAC-seq and scRNA-seq (lines 43–49) is imprecise and potentially misleading; differences in feature spaces are definitional, not inherently a "challenge," so the text should specify what the real computational challenges are.
>
> A2: We thank the reviewer for pointing out this imprecision and agree that modality-specific feature spaces are definitional rather than a challenge per se. Our intent was to highlight the computational difficulties that arise from this setting—namely high dimensionality, extreme sparsity (particularly in scATAC-seq), and the lack of one-to-one feature correspondences (e.g., one gene mapping to multiple peaks/regulatory regions). We have removed the vague term "substantial discrepancies" and to explicitly state "data sparsity, high dimensionality, and complex many-to-many regulatory relationships between genes and chromatin regions."
>
> W3: I don't actually believe your claim (line 55) that if you don't embed data into a shared space, then you "cannot fully exploit potentially complementary information across modalities." This is a very bold claim that requires substantial evidence. Indeed, I don't know how you could conclusively prove such a claim.
>
> A3: We thank the reviewer for this thoughtful comment. Our intent was to contrast with approaches that perform multimodal reconstruction without alignment, as discussed in the Related Work (lines 102–107). To avoid overstatement, we have revised line 55 to the following, more precise wording: "while effective for reconstruction, may have limitations for tasks requiring direct cross-modal comparison, such as cell-type querying or label transfer. Furthermore, by not creating a unified representation, they may not fully leverage complementary." This revision clarifies the task-dependent nature of the limitation and removes any implication of a universal or unprovable claim.
>
> Q4: I am not convinced that mean FOSCTTM is the most useful measure. Have you considered computing a p-value for improvement of the FOSCTTM?
>
> A4: A sign test is indeed a valid approach for evaluating per-cell improvements. We chose FOSCTTM as our primary alignment metric because it is a recognized and widely used deterministic measure in single-cell alignment, introduced by Singh et al. (2020) [1] and adopted by several state-of-the-art methods (e.g., GLUE, CLUE). Its chief advantage is an interpretable scalar score in [0, 1] that summarizes the overall quality of manifold alignment. We retain FOSCTTM to ensure fair comparison with prior work that reports the same metric, and we have clarified its formal definition in Appendix A.5.1.
>
> Q5. The critique at line 104 of reconstruction-based methods is unclear. Why are such methods “limited” for cross-modal comparison, querying, and label transfer when one can reconstruct one modality from the other and perform nearest-neighbor transfer without a shared latent space?
>
> A5: We do not claim reconstruction-based methods cannot do these tasks. The limitation is practical: they require an extra A→B inference per query (adding cost) and their accuracy is capped by reconstruction fidelity, so errors can propagate to downstream steps. A shared latent allows direct cross‑modal distances, querying, and label transfer in one space, which is typically more efficient and robust for symmetric comparisons. Revised sentence for line 104: "while powerful for reconstruction, they often do not produce a shared latent space. This can pose challenges for tasks like direct cross-modal querying or nearest-neighbor label transfer, which are more naturally performed within a unified embedding."
>
> [1] Singh, R. et al. Unsupervised manifold alignment for single-cell multi-omics data. In Proc. 11th ACM International Conference on Bioinformatics.

---

> ### Author Response · Authors · 2025-11-17
>
> W6: The text in lines 144–149 is unclear, including the distinction between "modality-specific features" and "semantic characteristics," and the meaning of "bounds of MI." Additionally, the sentence at lines 162–164 is ungrammatical, and the phrase "insufficient for effectively decoupling … in a directed manner" (lines 167–168) is confusing.
>
> A6: To address these issues comprehensively: (1) We now formally define the key terms in Section 3.1, grounded in our core assumption. Specifically, "semantic features" (Z_sem) denote modality-agnostic latent representations of shared cellular state, whereas "modality-specific features" (Z_spec) capture information unique to a given modality (e.g., modality-specific biological signals or technical noise) that we seek to disentangle prior to alignment.
>
> (2) The sentence at lines 162–164 has been rewritten to ensure grammatical correctness and improved readability.
>
> (3) The previously vague phrase "insufficient for effectively decoupling … in a directed manner" (lines 167–168) has been replaced with an explicit explanation: minimizing intra-modality mutual information I(Z_sem; Z_spec) alone is insufficient because, without guidance, the model lacks a criterion for allocating shared versus non-shared information. We therefore introduce an inter-modality alignment objective (maximizing I(Z_sem_RNA; Z_sem_ATAC)) to direct the model to place shared biological signal into Z_sem while retaining non-shared, modality-specific variation in Z_spec.
>
> (4) The ambiguous reference to "bounds of MI" has been clarified by explicitly stating we employ a tractable lower bound estimator for mutual information within the alignment and disentanglement losses.
>
> W7: The core assumption (line 184) should appear earlier to aid comprehension of feature definitions.
>
> A7: We moved the core assumption to Section 3.1 (Preliminaries) to establish early the distinction between Z_sem and Z_spec and improve conceptual continuity.
>
> W8: The dataset descriptions should cite prior papers that used them for benchmarking and specify the source of reported baseline results (own runs vs. extracted from literature).
>
> A8: The datasets are described in Appendix A.4, and all baseline results reported were reproduced by us. We explicitly state in Appendix A.4 that we executed all comparative methods ourselves.
>
> W9: I was surprised that all the talk about mutual a bound on MI ultimately seems to boil down to just doing an InfoNCE alignment loss.
>
> A9: We employ two distinct mutual information estimators: (1) For intra-modality disentanglement (minimizing I(Z_sem; Z_spec)) we use CLUB, a tractable upper bound estimator (Cheng et al., NeurIPS 2020); (2) For cross-modality alignment (maximizing I(Z_sem_RNA; Z_sem_ATAC)) we use InfoNCE, a standard lower bound ([2][3]). Using an upper bound for minimization and a lower bound for maximization is intentional and methodologically appropriate. We have revised Section 3.3 to state this explicitly and to justify the estimator choices with citations.
>
> W10: Typographical corrections are needed (line 192 "uses"→ "use"; line 270 "objection"→ "objective"). The sentence at line 293 about data being "hindered" by techniques/noise should be removed; just state the datasets. I do recognize that other papers in the literature use "reconstruction" to mean "translation."
>
> A10: We have corrected, and deleted the sentence at line 293. We acknowledge the terminological distinction between self-reconstruction (A→z→A′) and cross-modal translation (A→z→B′). We haven't implemented cross-modal translation, so we retain the term for comparability.
>
> W11: Did you compute the performance measures in Tables 1-4, or were some of these taken from previous publications? If the latter, did you use the same cross-validation splits?
>
> A11: We computed all results ourselves. Reported numbers are mean ± standard deviation over four random seeds. Baselines (e.g., MultiVI, GLUE) were rerun using their released code with default hyperparameters; all metrics and variances are from our runs.
>
> W12: How was train/test splitting done for each dataset?
>
> A12: As detailed in Appendix A.6, we use a 90%/10% train/validation split for all datasets, following GLUE and MultiVI. Alignment and related evaluations are conducted on the held-out validation split. For interpretability analyses (Appendix B.1.1; Tables 9 and 10), we use the full dataset, as these unsupervised/semi-supervised/self-supervised analyses do not consume labels and thus do not introduce label leakage.
>
> [2] Oord, A. V. D., Li, Y., & Vinyals, O. (2018). Representation learning with contrastive predictive coding. arXiv preprint arXiv:1807.03748.
>
> [3] Poole, B., Ozair, S., Van Den Oord, A., Alemi, A., & Tucker, G. (2019, May). On variational bounds of mutual information. In International conference on machine learning (pp. 5171-5180).

---

> ### Author Response · Authors · 2025-11-24
> **Addressing Clarity and Generalization**
>
> **Dear Reviewer JfMn,**
>
> Thank you again for your constructive feedback. As the discussion period draws to a close, we wanted to gently ensure that our responses and revisions have adequately addressed your concerns, particularly regarding the exposition clarity and model generalization.
>
> Based on your specific suggestions, we have made the following key improvements:
>
> **1. Evidence of Generalization (Your Q1):** You raised a crucial point about applying the model to independent, single-modality datasets. We have highlighted our **Cross-Dataset Zero-Shot Experiment (now detailed in Appendix B.1.4 and summarized in Section 4).** We trained scCMIA on mouse skin (SHARE-seq) and applied it directly to mouse cortex (SNARE-seq) without retraining. The results confirm that scCMIA generalizes well to new, unseen datasets, validating its practical utility.
>
> **2. Clarity and Formal Definitions (Your Q6 & Q7):** We completely restructured **Section 3.1 (Preliminary)**. We now formally define "semantic features" ($Z_{sem}$) and "modality-specific features" ($Z_{spec}$) before utilizing them. We also moved our core decoupling assumption to the beginning of this section to provide a clear logical foundation for our method.
>
> **3. Precise Problem Formulation (Your Major Comment):** We rewrote the **Introduction (Paragraph 2)** to remove ambiguity. We now explicitly distinguish between the challenges of "data sparsity/dimensionality" (replacing the vague "substantial discrepancies" per your Q2) and formally define the distinct tasks of "Cross-Modal Alignment" versus "Translation."
>
> **4. Refined Claims (Your Q3 & Q5):** We have revised the statements regarding the necessity of a shared latent space (Line 55) and the limitations of reconstruction-based methods (Line 104). We removed the "bold claims" and instead provided a nuanced explanation of why a unified space offers specific advantages for tasks like symmetric querying and label transfer.
>
> We believe these revisions significantly improve the manuscript's rigor and readability. We would appreciate it if you could let us know if any concerns remain.
>
> Sincerely,
>
> The Authors

---

> ### Author Response · Authors · 2025-11-27
> **To: Reviewer JfMn; Subject: Final Check: Clarity Revisions & Zero-Shot Generalization**
>
> **Dear Reviewer JfMn,**
>
> As the discussion period nears its end, we respectfully ask if you have had a chance to review our responses.
>
> We have specifically addressed your major concerns by:
>
> **1. Rewriting the Introduction and Methods** to provide the precise problem definition and formal terminology you requested.
>
> **2. Highlighting the Cross-Dataset Zero-Shot Experiment** (Appendix B.1.4) to demonstrate the model's generalization to independent, single-modality datasets.
>
> We believe these revisions significantly improve the manuscript's clarity and rigor. We would greatly appreciate your final feedback.
>
> Best regards,
>
> The Authors

---

### Author Response · Authors · 2025-11-30
**Rebuttal Summary: Consensus on Revisions, Resolved Concerns (uR8P, RAPg, JfMn, uq18), and Validated Innovations**

**Dear Area Chair,**

We thank the reviewers for their insightful feedback, which has driven us to significantly strengthen the empirical validation and clarity of scCMIA. We are pleased to report that **Reviewer RPAg and uR8P have explicitly increased their score**, and we have systematically resolved the concerns of the all reviewers.

We have addressed all raised concerns as follows:

**1. Novelty & Methodological Contribution (Response to uR8P, RAPg, uq18):** We clarified that scCMIA goes beyond a simple assembly of modules; it is a principled **Dual-Objective Framework** specifically engineered to resolve the inherent 'alignment-preservation trade-off' in single-cell omics.

**(1) Dual MI Framework:** By uniquely coupling the minimization of intra-modal MI (via CLUB) to strip technical noise with the maximization of inter-modal MI (via InfoNCE) for alignment, we achieve cleaner representations than standard VQ-VAEs.

**(2) Cross-Modal Unified Codebook (CrossVQ):** We introduced a novel constraint that forces diverse modalities into a shared discrete vocabulary. As demonstrated in our Appendix B.1.1 (Tables 9 and 10), this enables interpretable biological discovery (e.g., Regulatory Conservation Score), offering insights unavailable in continuous latent models.

**(3) Necessity of Decoupling (RAPg):** We clarified that decoupling is functionally essential to **prevent modality-specific noise from contaminating the shared alignment space**—a critical issue in sparse single-cell data. We provided rigorous empirical proofs (Table 4 & Figure 3) demonstrating that our strategy successfully isolates **"alignable" biological identity** (captured solely in $Z_{sem}$) from **"interfering" technical variation** (isolated in $Z_{spec}$). This quantitatively proves that decoupling is the key mechanism enabling scCMIA to achieve high-fidelity alignment without sacrificing reconstruction quality.

**2. Generalization & Zero-Shot Capabilities (Response to JfMn, RAPg):** Reviewers JfMn and RAPg asked about the model's applicability to new, independent datasets or uni-modal data.

We respectfully noted that a rigorous **Cross-Dataset Zero-Shot Experiment was already included in our original submission (Appendix B.1.4).** By training on mouse skin (SHARE-seq) and testing on mouse cortex (SNARE-seq) without fine-tuning, scCMIA maintained robust alignment accuracy. We have highlighted this result in the main text to demonstrate that scCMIA learns transferable biological knowledge and is fully applicable to **uni-modal data scenarios (answering RAPg)**.

**3. Complexity & Scalability (Response to uR8P):** To address concerns about dataset simplicity, we expanded validation to the **10x BMMC dataset (~69k cells, 13 batches)**. scCMIA achieved SOTA performance, **ranking (comprehensive scores) #1** in cell-type classification against leading methods like scBridge and MultiVI (Table 6, 7), demonstrating exceptional scalability.


**4. Reconstruction as Data Enhancement (Response to uq18):** Reviewer uq18 questioned the utility of reconstruction on noisy data.

**(1)** We clarified that our reconstruction module was explicitly designed for data enhancement and imputation. We highlighted our **Masking Experiments (Appendix B.1)**, where scCMIA successfully recovered ~96% of artificially masked scATAC-seq peaks. This proves the model functions as a high-fidelity **imputation engine** that restores lost biological signals from sparse data, refuting the "reconstruction is useless" critique.

**(2) Fairness:** We strictly followed **standard benchmarking protocols (Nature Methods, 2025)** for hyperparameter tuning and added **CITE-seq experiments** as requested.

**Conclusion:** scCMIA stands as a theoretically grounded and empirically robust solution that goes beyond standard integration. By resolving the fundamental **alignment-preservation trade-off** through our novel **Dual-Objective framework**, we enable **tangible biological discovery**—specifically, the quantification of regulatory conservation via our interpretable discrete codebook. With its **Zero-shot generalization** capabilities now clarified and its utility for high-fidelity imputation and and **large-scale downstream analysis** rigorously validated, we believe our work offers a distinct and valuable contribution to the field. We respectfully request the committee to consider these substantial improvements and the consensus among reviewers on the manuscript's enhanced quality.

Sincerely,

The Authors


Reference: Multitask benchmarking of single-cell multimodal omics integration methods. Nat Methods (2025).

---

### Note · Authors · 2026-01-27

I have read and agree with the venue's withdrawal policy on behalf of myself and my co-authors.

---

### Meta-Review · Area_Chair_F8Vt · 2026-01-06

**Summary:**

### Summary

The paper proposes scCMIA, a self-supervised framework for single-cell multimodal integration (primarily scRNA-seq + scATAC-seq) that aims to (i) align cells across modalities and (ii) enable cross-modal translation/retrieval, using a combination of contrastive MI-based alignment (InfoNCE) and disentanglement of semantic vs. modality-specific factors, with a shared discrete codebook (VQ) to improve interpretability. Experiments across multiple datasets and tasks suggest competitive performance versus established baselines.

### Strengths

- Combines disentangle-then-align objectives (intra-modal decoupling + inter-modal alignment) in a coherent framework.

- The shared discrete codebook offers a potentially interpretable representation and is empirically linked to cell-type signals.

- Empirical evaluation spans multiple tasks (alignment / label transfer / reconstruction) and several datasets.

### Weaknesses

- Limited technical novelty: the method is largely an integration of existing components (MI estimators/contrastive learning + VQ-style discretization), with insufficiently clear unique technical contribution.

- Clarity and positioning issues: reviewers found the problem formulation, terminology, and motivation confusing or overstated in places, weakening confidence in the paper’s framing.

- Evaluation concerns: questions remain about fairness/rigor (baseline tuning, split protocol clarity, metric choices and significance), and whether improvements are meaningfully separated from strong existing methods.

- Generalization scope: while additional experiments are mentioned, broader validation beyond common protocols and clearer evidence of robust gains remain limited/uncertain.


While the paper is well-motivated and empirically thorough, reviewers converged on concerns that the core contribution is incremental—a composition of known techniques with unclear novel insight. Presentation and evaluation details (problem definition, claims, splitting/fairness, and evidence of meaningful improvements) reduce confidence in the rigor and impact. Overall, the submission does not yet meet the bar for acceptance given the novelty + clarity + evaluation concerns.

**Reviewer Concerns:**

Rebuttal impact: addressed vs. outstanding

- JfMn
  - Addressed: generalization via cross-dataset zero-shot experiment; imprecise wording/overclaims; clarifications on metrics (FOSCTTM) and MI estimators; dataset/baseline provenance and splitting details; multiple clarity/typo issues.

  - Still outstanding: overall readability may remain a concern (many issues were structural); skepticism about whether shared-latent critiques are fully convincing; desire for stronger statistical testing on improvements.

- uq18

  - Addressed: added/clarified tasks (label transfer classification), variance reporting rationale, added CITE-seq experiment, clarified split vs. downstream test protocol, fixed table highlighting error.

  - Still outstanding: reviewer explicitly maintained concerns about statistical indistinguishability, fairness without tuning, and evaluation rigor; they stated issues were “not resolved” and kept the score.

- uR8P

  - Addressed: broader validation (incl. large-scale BMMC), added/clarified baselines (e.g., scButterfly; additional graph-based), clearer novelty narrative; reviewer increased score.

  - Still outstanding: core novelty concern may be only partially alleviated (still largely a “patchwork” perception, though softened).

- RPAg

  - Addressed: decoupling rationale/utility, unimodal usefulness, and novelty positioning as a field-specific adaptation; reviewer raised score after seeing revisions.

  - Still outstanding: residual novelty concern remains in principle (components largely existing), even if reviewer became satisfied given application value.

**Reviewer Scores:**

- JfMn: No change (would likely remain at 4).

- uq18: No change (explicitly said they will keep their score).

- uR8P: Would increase (they already indicated a score increase after rebuttal).

- RPAg: Would increase (they stated concerns resolved and raised their score).

---

### Decision · Program_Chairs · 2026-01-26

Reject